# OCEANBENCH:
# The Sea Surface Height Edition

**J. Emmanuel Johnson**[*]
CNRS UMR IGE
johnsonj@univ-grenoble-alpes.fr

**Quentin Febvre**[*]
IMT Atlantique
quentin.febvre@imt-atlantique.fr

**Anastasia Gorbunova**
CNRS UMR IGE

**Sammy Metref**
DATLAS

**Maxime Ballarotta**
CLS

**Julien Le Sommer**
CNRS UMR IGE

**Ronan Fablet**
IMT Atlantique

## Abstract

The ocean is a crucial component of the Earth's system. It profoundly influences human activities and plays a critical role in climate regulation. Our understanding has significantly improved over the last decades with the advent of satellite remote sensing data, allowing us to capture essential sea surface quantities over the globe, e.g., sea surface height (SSH). Despite their ever-increasing abundance, ocean satellite data presents challenges for information extraction due to their sparsity and irregular sampling, signal complexity, and noise. Machine learning (ML) techniques have demonstrated their capabilities in dealing with large-scale, complex signals. Therefore we see an opportunity for these ML models to harness the full extent of the information contained in ocean satellite data. However, data representation and relevant evaluation metrics can be *the* defining factors when determining the success of applied ML. The processing steps from the raw observation data to a ML-ready state and from model outputs to interpretable quantities require domain expertise, which can be a significant barrier to entry for ML researchers. In addition, imposing fixed processing steps, like committing to specific variables, regions, and geometries, will narrow the scope of ML models and their potential impact on real-world applications. **OceanBench** is a unifying framework that provides standardized processing steps that comply with domain-expert standards. It is designed with a flexible and pedagogical abstraction: it a) provides plug-and-play data and pre-configured pipelines for ML researchers to benchmark their models w.r.t. ML and domain-related baselines and b) provides a transparent and configurable framework for researchers to customize and extend the pipeline for their tasks. In this work, we demonstrate the `OceanBench` framework through a first edition dedicated to SSH interpolation challenges. We provide datasets and ML-ready benchmarking pipelines for the long-standing problem of interpolating observations from simulated ocean satellite data, multi-modal and multi-sensor fusion issues, and transfer-learning to real ocean satellite observations. The `OceanBench` framework is available at github.com/jejjohnson/oceanbench and the dataset registry is available at github.com/quentinf00/oceanbench-data-registry.

---

[*]These authors contributed equally to this work

# 1 Motivation

The ocean is vital to the Earth's system [28]. It plays a significant role in climate regulation regarding carbon [40] and heat uptake [87]. It is also a primary driver of human activities (e.g., maritime traffic and world trade, marine resources and services) [106, 93]. However, monitoring the ocean is a critical challenge: the ocean state can only partially be determined because most of the ocean consists of subsurface quantities that we cannot directly observe. Thus, to quantify even a fraction of the physical or biochemical ocean state, we must often rely only on surface quantities that we can monitor from space, drifting buoys, or autonomous devices. Satellite remote sensing, in particular, is one of the most effective ways of measuring essential sea surface quantities [2] such as sea surface height (SSH) [95], sea surface temperature (SST) [77], and ocean color (OC) [53]. While these variables characterize only a tiny portion of the ocean ecosystem, they present a gateway to many other derived physical quantities [93].

Although we can access observable sea surface quantities, they are generally irregularly and extremely sparsely sampled. For instance, satellite-derived SSH data has less than 5% coverage of the globe daily [95]. These sampling gaps make the characterization of ocean processes highly challenging for operational products and downstream tasks that depend on relevant gap-free variables. This has motivated a rich literature in geoscience over the last decades, mainly using geostatistical kriging methods [95, 102] and model-driven data assimilation schemes [55, 60]. Despite significant progress, these schemes often need to improve their ability to leverage available observation datasets' potential fully. This has naturally advocated for exploring data-driven approaches like shallow ML schemes [7, 6, 97, 71]. Very recently, deep learning schemes [116, 74, 9] have become appealing solutions to benefit from existing large-scale observation and simulation datasets and reach significant breakthroughs in the monitoring of upper ocean dynamics from scarcely and irregularly sampled observations. However, the heterogeneity and characteristics of the observation data present major challenges for effectively applying these methods beyond idealized case studies. A data source could have different variables, geometries, and noise levels, resulting in many domain-specific preprocessing procedures that can vastly change the solution outcome. Furthermore, the evaluation procedure of the methods and their effectiveness can be regionally-dependent as the physical phenomena vary in space and time, which adds another layer of complexity in convincing domain scientists of their trustworthiness. So the entire ML pipeline now requires a unified framework for dealing with heterogeneous data sources, different pre- and post-processing methodologies, and regionally-dependent evaluation procedures.

To address these challenges, we introduce **OceanBench**, a framework for co-designing machine-learning-driven high-level experiments from ocean observations. It consists of an end-to-end framework for piping data from its raw form to an ML-ready state and from model outputs to interpretable quantities. We regard `OceanBench` as a key facilitator for the uptake of MLOPs tools and research [66, 94] for ocean-related datasets and case studies. This first edition provides datasets and ML-ready benchmarking pipelines for SSH interpolation problems, an essential topic for the space oceanography community, related to ML communities dealing with issues like in-painting [111], denoising [99, 98], and super-resolution [107]. We expect `OceanBench` to facilitate new challenges to the applied machine learning community and contribute to meaningful ocean-relevant breakthroughs. The remainder of the paper is organized as follows: in §2, we outline some related work that was inspirational for this work; in §3, we formally outline `OceanBench` by highlighting the target audience, code structure, and problem scope; in §4, we outline the problem formulation of SSH interpolation and provide some insight into different tasks related to SSH interpolation where `OceanBench` could provide some helpful utility; and in §5 we give some concluding remarks while also informally inviting other researchers to help fill in the gaps.

# 2 Related Work

Machine learning applied to geosciences is becoming increasingly popular, but there are few examples of transparent pipelines involving observation data. After a thorough literature review, we have divided the field into three camps of ML applications that pertain to this work: 1) toy simulation datasets, 2) reanalysis datasets, and 3) observation datasets. We outline the literature for each of the three categories below.

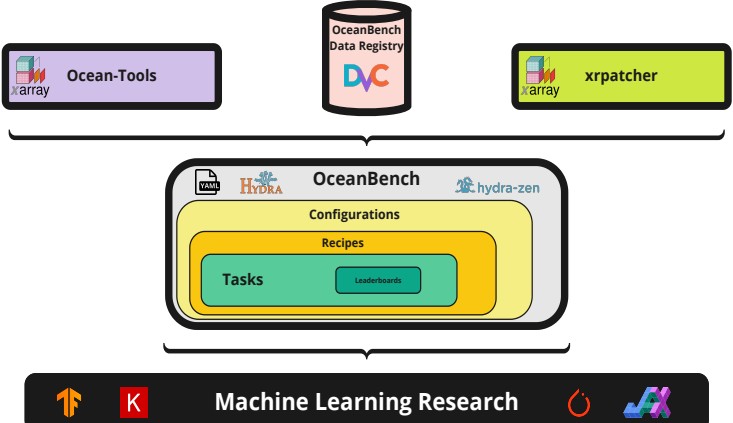

Figure 1: This figure showcases the `OceanBench` toolset. We have 1) `OceanBench-Data-Registry` which uses `DVC` to store and organize ML-ready datasets, 2) `Ocean-Tools` which features a suite of task-relevant geoprocessing functions with the `xarray`-backend, and 3) `xrpatcher` which can produce arbitrary subsets of `xarray` datastructures which nicely interface with dataloaders such as `PyTorch`. `OceanBench` provides an interface for ML researchers to parameterize arbitrary sequences of transformations to preprocess data from a domain-specific state to a ML-ready state.

**Toy Simulation Data**. One set of benchmarks focuses on learning surrogate models for well-defined but chaotic dynamical systems in the form of ordinary differential equations (ODEs) and partial differential equations (PDEs) and there are freely available code bases which implement different ODEs/PDEs [52, 96, 3, 64, 8, 103, 56, 85]. This is a great testing ground for simple toy problems that better mimic the structures we see in real-world observations. Working with simulated data is excellent because it is logistically simple and allows users to test their ideas on toy problems without increasing the complexity when dealing with real-world data. However, these are ultimately simple physical models that often do not reflect the authentic structures we see in real-world, observed data.

**Reanalysis Data**. This is assimilated data of real observations and model simulations. There are a few major platforms that host ocean reanalysis data like the Copernicus Marine Data Store [36, 33, 34, 37], the Climate Data Store [25], the BRAN2020 Model [26], and the NOAA platform [15]. However, to our knowledge, there is no standard ML-specific ocean-related tasks to accompany the data. On the atmospheric side, platforms like `WeatherBench` [86], `ClimateBench` [108], `ENS10` [10] were designed to assess short-term and medium-term forecasting using ML techniques with recent success of ML [69, 84] The clarity of the challenges set by the benchmark suites has inspired the idea of `OceanBench`, where we directly focus on problems dealing with ocean observation data.

**Observation Data**. These observation datasets (typically sparse) stem from satellite observations that measure surface variables or in-situ measurements that measure quantities within the water column. Some major platforms to host data include the Marine Data Store [32, 31], the Climate Data Store [23, 24, 22], ARGO [110], and the SOCAT platform [11]. However, it is more difficult to assess the efficacy of operational ML methods that have been trained only on observation data and, to our knowledge, there is no coherent ML benchmarking system for ocean state estimation. There has been significant effort by the *Ocean-Data-Challenge* Group[1] which provides an extensive suite of datasets and metrics for SSH interpolation. Their efforts heavily inspired our work, and we hope that `OceanBench` can build upon their work by adding cohesion and facilitating the ease of use for ML research and providing a high-level framework for providing ML-related data products.

## 3 OceanBench

### 3.1 Why OceanBench?

There is a high barrier to entry in working with ocean observations for researchers in applied machine learning as there are many processing steps for both the observation data and the domain-specific

---

[1]Ocean Data Challenge group: Freely associated scientist for oceanographic algorithm and product improvements (ocean-data-challenges.github.io)

evaluation procedures. `OceanBench` aims to lower the barrier to entry cost for ML researchers to make meaningful progress in the field of state prediction. We distribute a standardized, transparent, and flexible procedure for defining data and evaluation pipelines for data-intensive geoscience applications. Proposed examples and case studies provide a plug-and-play framework to benchmark novel ML schemes w.r.t. state-of-the-art, domain-specific ML baselines (see figure 1). In addition, we adopt a pedagogical abstraction that allows users to customize and extend the pipelines for their specific tasks. To our knowledge, no framework embeds processing steps for earth observation data in a manner compatible with MLOps abstractions and standards regarding reproducibility and evaluation procedures. Ultimately, we aim to facilitate the uptake of ML schemes to address ocean observation challenges and to bring new challenges to the ML community to extend additional ML tools and methods for irregularly-sampled and partially-observed high-dimensional space-time dynamics. The abstractions proposed here apply beyond ocean sciences and SSH interpolation to other geosciences with similar tasks that intersect with machine learning.

## 3.2 Code Structure

`OceanBench` is lightweight in terms of the core functionality. We keep the code base simple and focus more on how the user can combine each piece. We adopt a strict functional style because it is easier to maintain and combine sequential transformations. There are five features we would like to highlight about `OceanBench`: 1) Data availability and version control, 2) an agnostic suite of geoprocessing tools for `xarray` datasets that were aggregated from different sources, 3) Hydra integration to pipe sequential transformations, 4) a flexible multi-dimensional array generator from `xarray` datasets that are compatible with common deep learning (DL) frameworks, and 5) a JupyterBook [38] that offers library tutorials and demonstrates use-cases. In the following section, we highlight these components in more detail.

`OceanBench-Data-Registry`. The most important aspect is the public availability of the datasets. We aggregate all pre-curated datasets from other sources, e.g. the *Ocean-Data-Challenge* [13, 12], and organize them to be publicly available from a single source [2]. We also offer a few derived datasets which can be used for demonstrations and evaluation. Data is never static in a pipeline setting, as one can have many derived datasets which stem from numerous preprocessing choices. In fact, in research, we often work with derived datasets that have already been through some preliminary preprocessing methods. To facilitate the ever-changing nature of data, we use the Data Version Control (`DVC`) tool [67], which offers a git-like version control of the datasets.

`Ocean-Tools` [3]. The `Ocean-Tools` library uses a core suite of functions specific to processing geo-centric data. While a few particular functionalities vary from domain to domain, many operations are standard, e.g., data variable selections, filtering/smoothing, regridding, coordinate transformations, and standardization. We almost work exclusively with the `xarray` [58] framework because it is a coordinate-aware, flexible data structure. In addition, the geoscience community has an extensive suite of specialized packages that operate in the `xarray` framework to accomplish many different tasks. Almost all `Ocean-Tools` toolsets are exclusively within the `xarray` framework to maintain compatibility with a large suite of tools already available from the community.

**Hydra Integration**. As discussed above, many specific packages accomplish many different tasks. However, what needs to be added is the flexibility to mix and match these operations as the users see fit. `Hydra` [112] and `Hydra-Zen` [92] provide a configurable way to aggregate and *pipe* many sequential operations together. It also maintains readability, robustness, and flexibility through the use of `.yaml` files which explicitly highlights the function used, the function parameters chosen, and the sequence of operations performed. In the ML software stack, `Hydra` is often used to manage the model, optimizer, and loss configurations which helps the user experiment with different options. We apply this same concept in preprocessing, geoprocessing, and evaluation steps, often more important than the model configuration in geoscience-related tasks.

`XRPatcher` [4]. Every machine learning pipeline will inevitably require moving data from the geo-specific data structure to a multi-dimensional array easily digestible for ML models. A rather underrated, yet critical, feature of ML frameworks such as `PyTorch` [83] (`Lightning` [45]) and

---

[2]Available at: quentinf00/oceanbench-data-registry
[3]Available at: jejjohnson/ocn-tools
[4]Available at: jejjohnson/xrpatcher

`TensorFlow` [1] (`Keras` [30]) is the abstraction of the dataset, dataloader, datamodules, and data pipelines. In applied ML in geosciences, the data pipelines are often more important than the actual model [89]. The user can control the *patch*-size and the *stride*-step, which can generate arbitrary coordinate-aware items directly from the `xarray` data structure. In addition, `XRPatcher` provides a way to reconstruct the fields from an arbitrary patch configuration. This robust reconstruction step is convenient to extend the ML inference step where one can reconstruct entire fields of arbitrary dimensions beyond the training configuration, e.g., to account for the border effects within the field (see appendix E) or to reconstruct quantities in specific regions or globally.

**JupyterBook**. Building a set of tools is relatively straightforward; however, ensuring that it sees a broader adoption across a multi-disciplinary community is much more challenging. We invested heavily in showing use cases that appeal to different users with the `JupyterBook` platform [38]. Code with context is imperative for domain and ML experts as we need to explain and justify each component and give many examples of how they can be used in other situations. Thus, we have paid special attention to providing an extensive suite of tutorials, and we also highlight use cases for how one can effectively use the tools.

### 3.3 Problem Scope

There are many problems that are of great interest the ocean community [29] but we limit the scope to state estimation problems [21]. Under this scope, there are research questions that are relevant to operational centers which are responsible for generating the vast majority of global ocean state maps [36, 34, 33, 37] that are subsequently used for many downstream tasks [93]. For example: how can we effectively use heterogeneous observations to predict the ocean state on the sea surface [55, 62, 102, 44, 14, 77]; how can we incorporate prior physics knowledge into our predictions of ocean state trajectories [55, 29, 93]; and how can we use the current ocean state at time $T$ to predict the future ocean state at time $T + \tau$ [42, 86, 16]. In the same vain, there are more research questions that are of interest to the academic modeling community. For example: is simulated or reanalysis data more effective for learning ML emulators that replace expensive ocean models [49, 114]; what metrics are more effective for assessing our ability to mimic ocean dynamics [75, 48]; and how much model error can we characterize when learning from observations [18, 68].

We have cited many potential applications of how ML can be applied to tackle the state estimation problem. However, to our knowledge there is no publicly available, standardized benchmark system that is caters to ML-research standards. We believe that, irrespective of the questions posed above and the data we access, there are many logistical similarities for each of the problem formulations where we can start to set standards for a subset of tasks like interpolation or forecasting. On the front-end, we need a way to select regions, periods, variables, and a valid train-test split (see sec. D.1). On the back-end, we need a way to transform the predictions into more meaningful variables with appropriate metrics for validation (see sec. D.2 and D.3). `OceanBench` was designed to be an agnostic tool that is extensible to the types of datasets, processing techniques and metrics needed for working with a specific class of Ocean-related datasets. We strongly feel that a suite like this is the first step in designing task-specific benchmarks within the ocean community that is compatible with ML standards. In the remainder of the paper, we will demonstrate how `OceanBench` can be configured to facilite a ML-ready data challenge involving our first edition to demonstrate `OceanBench`'s applicability: sea surface height interpolation.

## 4 *Sea Surface Height Edition*

Sea surface height (SSH) is one of the most critical, observable quantities when determining the ocean state. It is widely used to study ocean dynamics and the adverse impact on global climate and human activities [78]. SSH enables us to track phenomena such as currents and eddies [78, 27, 82], which leads to a better quantification of the transport of energy, heat, and salt. In addition, SSH helps us quantify sea level rise at regional and global scales [4, 39], which is used for operational monitoring of the marine environment [106]. Furthermore, SSH characterization provides a plethora of data products that downstream tasks can use for many other applications [79, 20]. Due to the irregular sampling delivered by satellite altimeter, state-of-the-art operational methods using optimal interpolation schemes [95, 102] or model-driven data assimilation [7, 6, 71, 97] fail to fully retrieve SSH dynamics at fine scales below 100-200km on a global or regional scale, so improving the

space-time resolution of SSH fields has been a critical challenge in ocean science. Beyond some technological developments [51], recent studies support the critical role of ML-based schemes in overcoming the current limitations of the operational systems [14, 55, 116] . The rest of this section gives an overview of the general problem definition for SSH interpolation, followed by a brief ontology for ML approaches to address the problem. We also give an overview of some experimental designs and datasets with a demonstration of metrics and plots generated by the `OceanBench` platform.

## 4.1 Problem Definition

We are dealing with satellite observations, so we are interested in the domain across the Earth's surface. Let us define the Earth's domain by some spatial coordinates, $\mathbf{x} = [\text{Longitude}, \text{Latitude}]^\top \in \mathbb{R}^{D_s}$, and temporal coordinates, $t = [\text{Time}] \in \mathbb{R}^+$, where $D_s$ is the dimensionality of the coordinate vector. We can define some spatial (sub-)domain, $\Omega \subseteq \mathbb{R}^{D_s}$, and a temporal (sub-)domain, $\mathcal{T} \subseteq \mathbb{R}^+$. This domain could be the entire globe for 10 years or a small region within the North Atlantic for 1 year.

$$\text{Spatial Coordinates}: \qquad \mathbf{x} \in \Omega \subseteq \mathbb{R}^{D_s} \qquad (1)$$

$$\text{Temporal Coordinates}: \qquad t \in \mathcal{T} \subseteq \mathbb{R}^+. \qquad (2)$$

In this case $D_s = 2$ because we only have a two coordinates, however we can do some coordinate transformations like spherical to Cartesian. Likewise, we can do some coordinate transformation for the temporal coordinates like cyclic transformations or sinusoidal embeddings [105]. We have two fields of interest from these spatiotemporal coordinates: the state and the observations.

$$\text{State}: \qquad \boldsymbol{u}(\mathbf{x}, t) : \Omega \times \mathcal{T} \to \mathbb{R}^{D_u} \qquad (3)$$

$$\text{Observations}: \qquad \boldsymbol{y}_{obs}(\mathbf{x}, t) : \Omega \times \mathcal{T} \to \mathbb{R}^{D_{obs}} \qquad (4)$$

The state domain, $u \in \mathcal{U}$, is a scalar or vector-valued field of size $D_u$ which is typically the quantity of interest and the observation domain, $y_{obs} \in \mathcal{Y}_{obs}$, is the observable quantity which is also a scalar or vector-valued field of size $D_{obs}$. Now, we make the assumption that we have an operator $\mathcal{H}$ that transforms the field from the state space, $\boldsymbol{u}$, to the observation space, $\boldsymbol{y}_{obs}$.

$$\boldsymbol{y}_{obs}(\mathbf{x}, t) = \mathcal{H}\left(\boldsymbol{u}(\mathbf{x}, t), t, \boldsymbol{\varepsilon}, \boldsymbol{\mu}\right) \qquad (5)$$

This equation is the continuous function defined over the entire spatiotemporal domain. The operator, $\mathcal{H}(\cdot)$, is flexible and problem dependent. For example, in a some discretized setting there are 0's wherever there are no observations, and 1's wherever there are observations, and in other discretized settings it takes a weighted average of the neighboring pixels. We also include a generic noise function, $\boldsymbol{\varepsilon}(\mathbf{x}, t)$. This could stem from a distribution, it could stationary noise operator, $\boldsymbol{\varepsilon}(\mathbf{x})$, or it could be constant in space but vary with Time, $\boldsymbol{\varepsilon}(t)$. We also include a control parameter, $\boldsymbol{\mu}$, representing any external factors or latent variables that could connect the state vector to the observation vector, e.g., sea surface temperature. Our quantity of interest is SSH, $\eta$, a scalar-valued field defined everywhere on the domain. In our application, we assume that the SSH we observe from satellite altimeters, $\eta_{obs}$, is the same as the SSH state, except it could be missing for some coordinates due to incomplete coverage from the satellite. So our transformation is defined as follows:

$$\boldsymbol{\eta}_{obs}(\mathbf{x}, t) = \mathcal{H}\left(\boldsymbol{\eta}(\mathbf{x}, t), t, \boldsymbol{\varepsilon}, \boldsymbol{\mu}\right) \qquad (6)$$

In practice, the satellite providers have a reasonable estimation of the amount of structured noise level we can expect from the satellite altimetry data; however, unresolved noise could still be present. Finally, we are interested in finding some model, $\mathcal{M}$, that maps the SSH we observe to the true SSH given by

$$\mathcal{M} : \boldsymbol{\eta}_{obs}(\mathbf{x}, t, \boldsymbol{\mu}) \to \boldsymbol{\eta}(\mathbf{x}, t), \qquad (7)$$

which is essentially an inverse problem that maps the observations to the state. One could think of it as trying to find the inverse operator, $\mathcal{M} = \mathcal{H}^{-1}$, but this could be some other arbitrary operator.

## 4.2 Machine Learning Model Ontology

In general, we are interested in finding some parameterized operator, $\mathcal{M}_{\boldsymbol{\theta}}$, that maps the incomplete SSH field to the complete SSH field

$$\mathcal{M}_{\boldsymbol{\theta}} : \boldsymbol{\eta}_{obs}(\mathbf{x}, t, \boldsymbol{\mu}) \to \boldsymbol{\eta}(\mathbf{x}, t), \qquad (8)$$

whereby we learn the parameters from data. The two main tasks we can define from this problem setup are 1) interpolation and 2) extrapolation. We define *interpolation* as the case when the boundaries of the inferred state domain lie within a predefined shape for the boundaries of the spatiotemporal observation domain. For example, the shape of the spatial domain could be a line, box, or sphere, and the shape of the temporal domain could be a positive real number line. We define *extrapolation* as the case where the boundaries of the inferred state domain are outside the boundaries of the spatiotemporal observation domain. In this case, the inferred state domain could be outside of either domain or both. A prevalent specific case of extrapolation is *hindcasting* or *forecasting*, where the inferred state domain lies within the spatial observation domain's boundaries but outside of the temporal observation domain's. In the rest of this paper, we will look exclusively at the interpolation problem. However, we refer the reader to appendix F for a more detailed look at other subtasks that can arise.

From a ML point of view, we can explore various ways to define the operator in equation (7). We may distinguish three main categories: (i) coordinate-based methods that learn a parameterized continuous function to map the domain coordinates to the scalar values, (ii) the explicit mapping of the state from the observation, (iii) implicit methods defined as the solution of an optimization problem. The first category comprises of kriging approaches, which have been used operationally with historical success [113, 95]. Beyond such covariance-based approaches, recent contributions explore more complex trainable functional models [72], basis functions [102], and neural networks [62]. The second category of schemes bypasses the physical modeling aspect and amortizes the prediction directly using state-of-the-art neural architectures such as UNets and ConvLSTMs [116, 74, 9]. This category may straightforwardly benefit from available auxiliary observations [23, 24, 22] to state the interpolation problem as a super-resolution [107] or image-to-image translation problem [81, 59]. The third category relates to inverse problem formulations and associated deep learning schemes, for example deep unfolding methods and plug-and-play priors [115]. Interestingly, recent contributions explore novel neural schemes which combine data assimilation formulations [21] and learned optimizer strategies [14, 44]. We provide a more detailed ontology of methods used for interpolation problems in appendix G. We consider at least one baseline approach from each category for each data challenge described in section 4.4. While all these methods have pros and cons, we expect the OceanBench platform to showcase to new experimental evidence and understanding regarding their applicability to SSH interpolation problems.

## 4.3 Experimental Design

Table 1: This table gives a brief overview of the datasets provided to complete the data challenges listed in 4.4 and A. Note that the OSSE datasets are all gridded products whereas the OSE NADIR is an alongtrack product. See figure 2 for an example of the OSSE NEMO Simulations for SSH and SST and pseudo-observations for NADIR & SWOT.

|  | OSSE | OSSE NADIR + SWOT | OSSE SST | OSE NADIR |
|---|---|---|---|---|
| Data Type | Simulations | Pseudo-Observations | Simulations | Observations |
| Source | NEMO [5] | NEMO [5] | NEMO [5] | Altimetry [32] |
| Region | GulfStream | GulfStream | GulfStream | GulfStream |
| Domain Size | $10 \times 10°$ | $10 \times 10°$ | $10 \times 10°$ | $10 \times 10°$ |
| Longitude Extent | $[-65°, -55°]$ | $[-65°, -55°]$ | $[-65°, -55°]$ | $[-65°, -55°]$ |
| Latitude Extent | $[33°, 43°]$ | $[33°, 43°]$ | $[33°, 43°]$ | $[33°, 43°]$ |
| Resolution | $0.05° \times 0.05°$ | $0.05° \times 0.05°$ | $0.05° \times 0.05°$ | 7 km |
| Grid Size | $200 \times 200$ | $200 \times 200$ | $200 \times 200$ | N/A |
| Num Datapoints | $\sim$14.6M | $\sim$14.6M | $\sim$14.6M | $\sim$1.6M |
| Period Start | 2012-10-01 | 2012-10-01 | 2012-10-01 | 2016-12-01 |
| Period End | 2013-09-30 | 2013-09-30 | 2013-09-30 | 2018-01-31 |
| Frequency | Daily | Daily | Daily | 1 Hz |

The availability of multi-year simulation and observation datasets naturally advocates for the design of synthetic (or twin) experiments, referred to as observing system simulation experiments (OSSE), and of real-world experiments, referred to as observing system experiments (OSE). We outline these two experimental setups below.

**Observing System Simulation Experiments (OSSE).** A staple and groundtruthed experimental setup uses a reference simulation dataset to simulate the conditions we can expect from actual satellite observations. This setup allows researchers and operational centers to create a fully-fledged pipeline that mirrors the real-world experimental setting. An ocean model simulation is deployed over a specified spatial domain and period, and a satellite observation simulator is deployed to simulate satellite observations over the same domain and period. This OSSE setup has primarily been considered for performance evaluation, as one can assess a reconstruction performance over the entire space-time domain. It also provides the basis for the implementation of classic supervised learning strategies [9, 74, 116]. The domain expert can vary the experimental conditions depending on the research question. For example, one could specify a region based on the expected dynamical regime [12] or add a certain noise level to the observation tracks based on the satellite specifications. The biggest downside to OSSE experiments is that we train models exclusively with ocean simulations which could produce models that fail to generalize to the actual ocean state. Furthermore, the simulations are often quite expensive, which prevents the community from having high spatial resolution over very long periods, which would be essential to capture as many dynamical regimes as possible.

**Observing System Experiments (OSE).** As more observations have become available over the past few decades, we can also design experiments using real data. This involves aggregating as many observations from real ocean altimetry satellites as possible with some specific independent subset left out for evaluation purposes. A major downside to OSE experiments is that the sparsity and spatial coverage of the observations narrow the possible scope of performance metrics and make it very challenging to learn directly from observation datasets. The current standard altimetry data are high resolution but cover a tiny area. As such, it can only inform fine-scale SSH patterns in the along-track satellite direction and cannot explicitly reveal two-dimensional patterns. Despite these drawbacks, it provides a quantitative evaluation of the generalizability of the ML methods concerning the true ocean state.

## 4.4   Data Challenges

We rely on existing OSSE and OSE experiments for SSH interpolation designed by domain experts [13, 12] and recast them into `OceanBench` framework to deliver a ML-ready benchmarking suites. The selected data challenges for this first edition address SSH interpolation for a 1000km×1000km Gulfstream region. We briefly outline them below.

**Experiment I (*OSSE NADIR*)** addresses SSH interpolation using NADIR altimetry tracks which are very fine, thin ocean satellite observations (see Figure 2). It relies on an OSSE using high-resolution ($1/60°$ resolution) ocean simulations generated by the NEMO model over one year with a whole field every day.

**Experiment II (*OSSE SWOT*)** addresses SSH interpolation using jointly NADIR and SWOT altimetry data where we complement the **OSSE NADIR** configuration with simulated SWOT observations. SWOT is a new satellite altimetry mission with a much higher spatial coverage but a much lower temporal resolution as illustrated in Figure 2. The higher spatial resolution allows us to see structures at a smaller resolution but at the cost of a massive influx of observations (over ×100).

**Experiment III (*OSSE SST*)** addresses SSH interpolation using altimetry and SST satellite data jointly. We complement the **OSSE SWOT** challenge with simulated SST observations. Satellite-derived SST observations are more abundantly available in natural operational settings than SSH at a finer resolution, and structures have visible similarities [51, 55]. So this challenge allows for methods to take advantage of multi-modal learning [44, 116].

**Experiment IV (*OSE NADIR*)** addresses SSH interpolation for real NADIR altimetry data. In contrast to the three OSSE data challenges, it only looks at actual observations aggregated from the currently available ocean altimetry data from actual satellites. It involves a similar space-time sampling as Experiment (**OSSE NADIR**) to evaluate the generalization of ML methods trained in Experiment I to real altimetry data. The training problem's complexity increases significantly due to the reference dataset's sparsity compared with the **OSSE NADIR** dataset. One may also explore transfer learning or fine-tuning strategies from the available OSSE dataset.

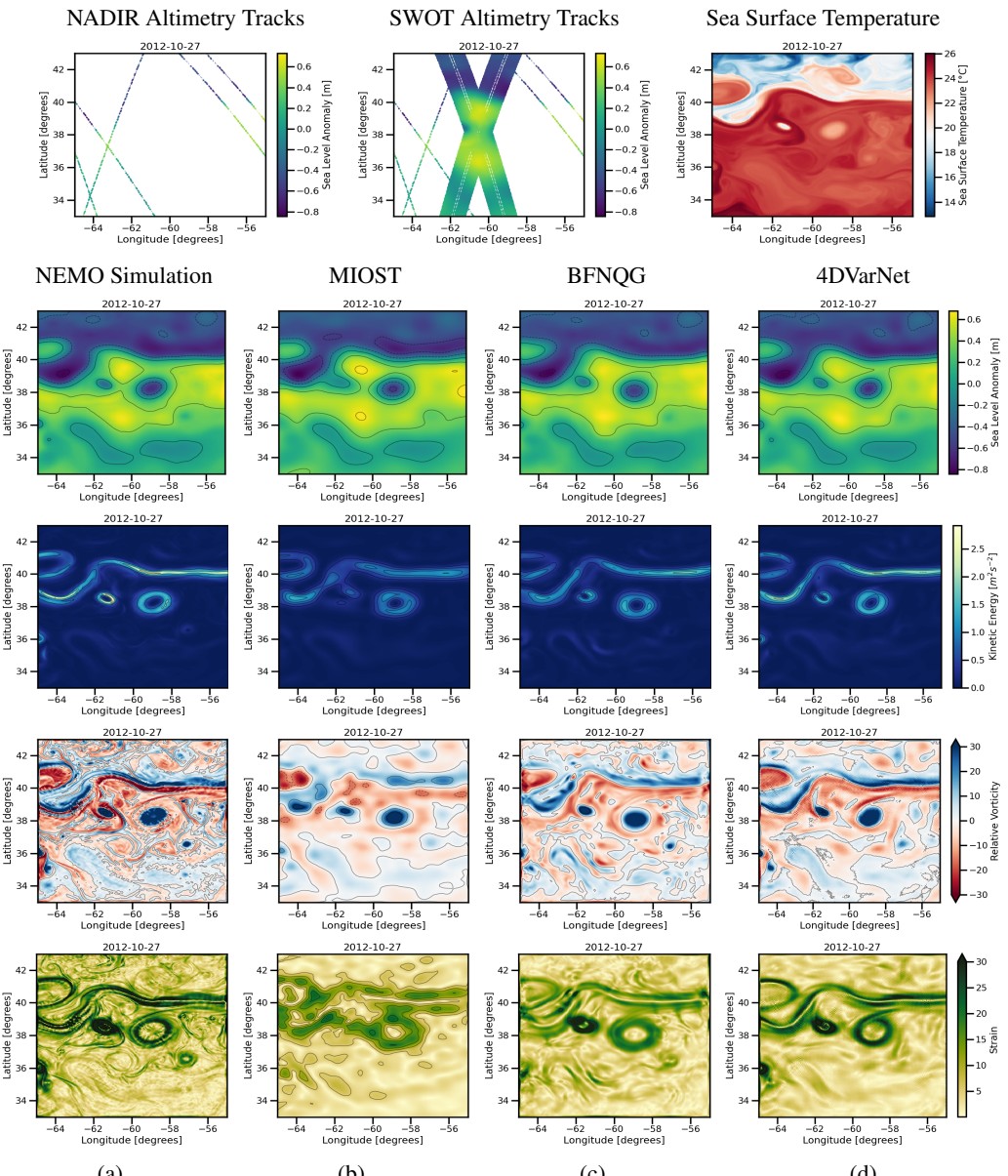

Figure 2: A snapshot at $27^{th}$ October, 2012 of the sea level anomaly (SLA) from the NEMO simulation for the OSSE experiment outlined in section 4.3. The top row showcases the aggregated NADIR altimetry tracks and the aggregated SWOT altimetry tracks (12 hours before and 12 hours after) as well as the SST from the NEMO simulation. Each subsequent row showcases the following physical variables found in appendix B: (a) Sea Level Anomaly, (b) Kinetic Energy, (c) Relative Vorticity, and (d) Strain. Each column in the subsequent rows showcase the following reconstructed field from the NEMO simulation found in columrn (a): (b) MIOST [102], (c) BFN-QG [55], and (d) 4DVarNet [14].

## 4.5  `OceanBench` **Pipelines**

For the four data challenges presented in the previous section, we used `OceanBench` pipelines to deliver a ML-ready benchmarking framework. We used the `hydra` and the geoprocessing tools outlined in section 3.2 with specialized routines for regridding the ocean satellite data to a uniformly gridded product and vice versa when necessary. Appendix D showcases an example of the hydra integration for the preprocessing pipeline. A key feature is the creation of a custom patcher for the appropriate geophysical variables using our `XRPatcher` tool, which is later integrated into custom datasets and dataloaders for the appropriate model architecture, e.g., coordinate-based or grid-based.

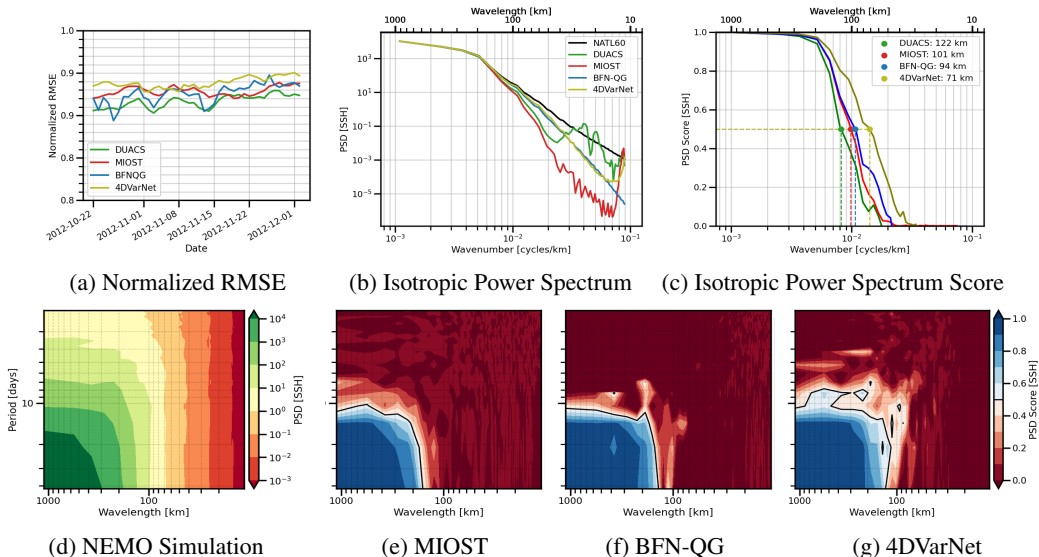

(a) Normalized RMSE     (b) Isotropic Power Spectrum     (c) Isotropic Power Spectrum Score

(d) NEMO Simulation    (e) MIOST    (f) BFN-QG    (g) 4DVarNet

Figure 3: This figure showcases some statistics for evaluation of the SSH field reconstructions for the OSSE NADIR experiment outlined in section 4. Subfigure (a) showcases the normalized root mean squared error (nRMSE), (b) showcases the isotropic power spectrum decomposition (PSD), (c) showcases isotropic PSD scores. The bottom row showcases the space-time PSD for the NEMO simulation (subfigure (d)) and the PSD scores for three reconstruction models: (e) the MIOST model [102], (f) the BFN-QG model [55], and (g) the 4DVarNet model [14].

Table 2: This table highlights some of the results for the **OSSE NADIR** experiment outlined in section 4.4 and appendix A. This table highlights the performance statistically in the real and spectral space; the normalized RMSE score for the real space and the minimum spatial and temporal scales resolved in the spectral domain. For more information about the class of models displayed and class of metrics, see appendix G and appendix C respectively. We only showcase the model performance on the alongtrack NADIR data available. For the extended table for each of the challenges, see Table 3.

| Experiment | Algorithm | Algorithm Class | nRMSE Score | $\lambda_{\mathbf{x}}$ [km] | $\lambda_t$ [days] |
|---|---|---|---|---|---|
| OSSE NADIR | OI [95] | Coordinate-Based | $0.92 \pm 0.01$ | 175 | 10.8 |
| OSSE NADIR | MIOST [102] | Coordinate-Based | $0.93 \pm 0.01$ | 157 | 10.1 |
| OSSE NADIR | BFNQG [55] | Hybrid Model | $0.93 \pm 0.01$ | 139 | 10.6 |
| OSSE NADIR | 4DVarNet [14] | Bi-Level Opt. | $0.95 \pm 0.01$ | 117 | 7.7 |

We provide an example snippet of how this can be done easily in section E. `OceanBench` also features some tools specific to the analysis of SSH. For example, physically-interpretable variables like geostrophic currents and relative vorticity, which can be derived from first-order and second-order derivatives of the SSH, are essential for assessing the quality of the reconstructions generated by the models. Figure 2 showcases some fields of the most common physical variables used in the oceanography literature for the SSH-based analysis of sea surface dynamics. For more details regarding the nature of the physical variables, see appendix B.

Regarding the evaluation framework, we include domain-relevant performance metrics beyond the standard ML loss and accuracy functions. They account for the sampling patterns of the evaluation data. Spectral analytics are widely used in geoscience [55], and here, we consider spectral scores computed as the minimum spatial and temporal scales resolved by the reconstruction methods proposed in [55]. For example, figure 3 showcases how `OceanBench` generated the isotropic power spectrum and score and the space-time power spectrum decomposition and score. Table 2 outlines some standard and domain-specific scores for the experiments outlined in section 4.3. We give a more detailed description of the rationale and construction of the power-spectrum-specific metrics in

appendix C. In terms of baselines, we report for each data challenge the performance of at least one approach for each of the category outlined in Section 4.2.

## 5 Conclusions

The ocean community faces technological and algorithmic challenges to make the most of available observation and simulation datasets. In this context, recent studies evidence the critical role of ML schemes in reaching breakthroughs in our ability to monitor ocean dynamics for various space-time scales and processes. Nevertheless, domain-specific preprocessing steps and evaluation procedures slow down the uptake of ML toward real-world applications. Our application of choice was SSH mapping which facilities the production of many crucial derived products that are used in many downstream tasks like subsequent modeling [93], ocean health monitoring [101, 73, 47] and maritime risk assessment [106].

Through `OceanBench` framework, we embed domain-level requirements into the MLOPs considerations by building a flexible framework that adds this into the hyperparameter considerations for ML models. We proposed four challenges towards a ML-ready benchmarking suite for ocean observation challenges. We outlined the inner workings `OceanBench` and demonstrated its usefulness by recreating some preprocessing and analysis pipelines from a few data challenges involving SSH interpolation. We firmly believe that the `OceanBench` platform is a crucial step to lowering the barrier of entry for new ML researchers interested in applying and developing their methods to relevant problems in the ocean sciences.

## Acknowledgments and Disclosure of Funding

This work was supported by the French National Research Agency (ANR), through projects number ANR-17- CE01-0009-01, ANR-19-CE46-0011 and ANR-19-CHIA-0016); by the French National Space Agency (CNES) through the SWOT Science Team program (projects MIDAS and DIEGO) and the OSTST program (project DUACS-HR); by the French National Centre for Scientific Research (CNRS) through the LEFE-MANU program (project IA-OAC). This project also received funding from the European Union's Horizon Europe research and innovation programme under the grant No 101093293 (EDITO-Model Lab project). This project benefited from HPC and GPU computing resources from GENCI-IDRIS (Grant 2021-101030).

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
