# OCEANBENCH: The Sea Surface Height Edition - Supplementary Material

## A   Data Challenges

In this section, we highlight some details that were omitted in section 4.4. This includes details about the simulation type, the data structures, and the training/evaluation periods.

### A.1   OSSE NADIR

The reference simulation is the *NATL60* simulation based on the NEMO model [5]. This particular simulation was run over an entire year without any tidal forcing. The simulation provides the outputs of SSH, SST, sea surface salinity (SSS) and the u,v velocities every 1 hour. For the purposes of this data challenge, the spatial domain is over the Gulfstream with a spatial domain of $[-65°, -55°]$ longitude and $[33°, 43°]$ latitude. The resolution of the original simulation is 1/60° resolution with hourly snapshots, and we consider a daily downsampled trajectory at 1/20° for the data challenge which results in a 365x200x200 spatio-temporal grid. This simulation resolves finescale dynamical processes (∼15km) which makes it a good test bed for creating an OSSE environment for mapping. The SSH observations include simulations of ocean satellite NADIR tracks. In particular, they are simulations of Topex-Poseidon, Jason 1, Geosat Follow-On, and Envisat. There is no observation error considered within the challenge. We use a the entire period from 2012-10-10 until 2013-09-30. A training period is only from 2013-01-02 to 2013-09-30 where the users can use the reference simulation as well as all available simulated observations. The evaluation period is from 2012-10-22 to 2012-12-02 (i.e. 41 days) which is considered decorrelated from the training period. During the evaluation period, the user cannot use the reference NATL60 simulation but they can use all available simulated observations. There is also a spin-up period allowance from 2012-10-01 where the user can also use all available simulated observations.

### A.2   OSSE SWOT & OSSE SST

For the OSSE SWOT and OSSE SST experiments, the reference simulation, domain, and evaluation period is the same as the OSSE NADIR experiment. However, the OSSE SWOT includes simulated observations of the novel KaRIN sensor recently deployed during the SWOT mission, the pseudo-observations were generated using the SWOT simulator [51]. This OSSE SST experiment allows the users to utilize the full fields of SST as inputs to help reconstruct the SSH field in conjunction with the NADIR and SWOT SSH observation. Because the SST comes from the same NATL60 simulation, the geometry characteristics SST and SSH are exactly the same.

### A.3   OSE NADIR

The OSE NADIR experiment only uses real observations aggregated from different altimeters. These SSH observations include observations from the SARAL/Altika, Jason 2, Jason 3, Sentinel 3A, Haiyang-2A and Cryosat-2 altimeters. The Cryosat-2 altimeter is used as the independent evaluation track used to assess the performance of the reconstructed SSH field.

### A.4   Results

We use `OceanBench` to generate maps of relevant quantities from the 4DVarNet method [14, 44]. Figure 4 showcases some demo maps for some key physical variables outlined in section B. We showcase the 4DVarNet method because it is the SOTA method that was applied to each of the data challenges. We can see that the addition of more information, i.e. NADIR -> SWOT -> SST, results in maps look more similar to the NEMO simulation in the OSSE challenges. It also produces sensible maps for the OSE challenge as well.

`OceanBench` also generated figure 5 which shows plots of the PSD and PSD scores of SSH for the different challenges. Again, as we increase the efficacy of the observations via SWOT and allow for more external factors like the SST, we get an improvement in the isotropic and spacetime PSD scores. In addition, we see that the PSD plots for the OSE task look very similar to the OSE challenges.

Lastly, we used `OceanBench` to generate a leaderboard of metrics for a diverse set of algorithms where the maps were available online. Table 3 displays all of the key metrics outlined in section C

including the normalized RMSE and various spectral scores which are appropriate for the challenge. We see that as the complexity of the method increases, the metrics improve. In addition, the methods that involve end-to-end learning perform the best overall, i.e. 4DVarNet.

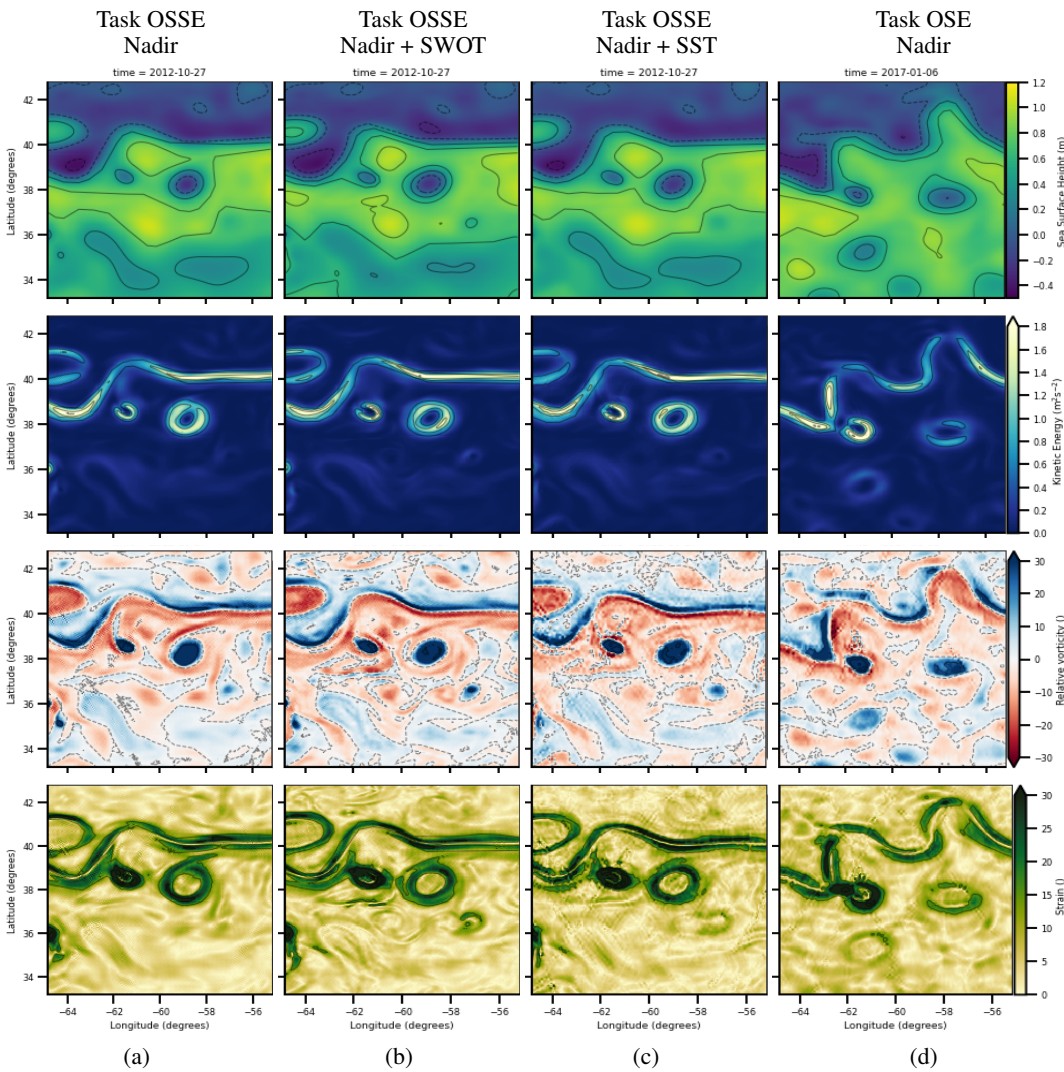

(a)  (b)  (c)  (d)

Figure 4: Reconstructed quantities by the 4dVarNet method for each of the four tasks. Each row showcases the following physical variables found in appendix B: (a) Sea Surface Height, (b) Kinetic Energy, (c) Relative Vorticity, and (d) Strain. Each column showcase the reconstructed from the tasks (a) OSSE using only Nadir tracks: (b) OSSE using Nadir tracks and SWOT swath, (c) Multimodal using Nadir tracks and sea surface temperature, and (d) Reconstruction using real nadir altimetry tracks.

## A.5  Datasets

In Table 4, we showcase all of the available datasets in our[5] for the challenges listed in the above section. The license for the datasets listed in the registry are under the CCA 4.0 International License.

---

[5]Available at: OceanBench Data Registry

Table 3: This table showcases all of the summary statistics for some methods for each of the data challenges listed in section 4.4 and A. The summary statistics shown are the normalized RMSE and the effective resolution in the spectral domain. The spectral metrics for the effective resolution that were outlined in section C are: i) $\lambda_a$ is the spatial score for the alongtrack PSD score, ii) $\lambda_r$ is the spatial score for the isotropic PSD, iii) $\lambda_x$ is the spatial score for space-time PSD score, and iv) $\lambda_t$ is the temporal score for the space-time PSD score.

| Experiment | Algorithm | nRMSE Score | Effective Resolution | | | |
| | | | $\lambda_a$ [km] | $\lambda_r$ [km] | $\lambda_\mathbf{x}$ [km] | $\lambda_t$ [days] |
|---|---|---|---|---|---|---|
| OSSE NADIR | OI | 0.92 | - | 123 | 174 | 10.8 |
| OSSE NADIR | MIOST | 0.93 | - | 100 | 157 | 10.1 |
| OSSE NADIR | BFNQG | 0.93 | - | 88 | 139 | 10.4 |
| OSSE NADIR | 4DVarNet | **0.94** | - | **65** | **117** | **7.7** |
| OSSE SWOT | OI | 0.92 | - | 106 | 139 | 11.7 |
| OSSE SWOT | MIOST | 0.94 | - | 88 | 131 | 10.1 |
| OSSE SWOT | BFNQG | 0.94 | - | 64 | 118 | 36.5 |
| OSSE SWOT | 4DVarNet | **0.96** | - | **47** | **77** | **5.6** |
| OSSE SST | Musti | 0.95 | - | 46 | 138 | 4.1 |
| OSSE SST | 4DVarNet | **0.96** | - | **46** | **87** | **3.7** |
| OSE NADIR | OI | 0.88 | 151 | - | - | - |
| OSE NADIR | MIOST | 0.90 | 135 | - | - | - |
| OSE NADIR | BFNQG | 0.88 | 122 | - | - | - |
| OSE NADIR | ConvLSTM | 0.89 | 113 | - | - | - |
| OSE NADIR | 4DVarNet | **0.91** | **98** | - | - | - |

Table 4: This table gives an extended overview of the datasets provided to complete the data challenges listed in 4.4 and A. The OSSE SST and SSH are outputs from come from the free run NEMO model [5]. The OSSE NADIR and SWOT are pseudo-observations generated from the NEMO simulation. We provide the original simulated satellite tracks as well as a gridded version at the same resolution as the simulation.

| | OSSE SSH | OSSE SSH NADIR | | OSSE SSH SWOT | | OSSE SST | OSE SSH NADIR |
|---|---|---|---|---|---|---|---|
| Data Structure | Gridded | AlongTrack | Gridded | AlongTrack | Gridded | Gridded | AlongTrack |
| Source | NEMO [5] | NEMO [5] | NEMO [5] | NEMO [5] | NEMO [5] | NEMO [5] | Altimetry [32] |
| Region | GulfStream | GulfStream | GulfStream | GulfStream | GulfStream | GulfStream | GulfStream |
| Domain Size [degrees] | $10 \times 10°$ | $10 \times 10°$ | $10 \times 10°$ | $10 \times 10°$ | $10 \times 10°$ | $10 \times 10°$ | $10 \times 10°$ |
| Domain Size [km] | $1,100 \times 1,100$ | $1,100 \times 1,100$ | $1,100 \times 1,100$ | $1,100 \times 1,100$ | $1,100 \times 1,100$ | $1,100 \times 1,100$ | $1,100 \times 1,100$ |
| Longitude Extent | $[-65°, -55°]$ | $[-65°, -55°]$ | $[-65°, -55°]$ | $[-65°, -55°]$ | $[-65°, -55°]$ | $[-65°, -55°]$ | $[-65°, -55°]$ |
| Latitude Extent | $[33°, 43°]$ | $[33°, 43°]$ | $[33°, 43°]$ | $[33°, 43°]$ | $[33°, 43°]$ | $[33°, 43°]$ | $[33°, 43°]$ |
| Resolution [degrees] | $0.05° \times 0.05°$ | N/A | $0.05° \times 0.05°$ | N/A | $0.05° \times 0.05°$ | $0.05° \times 0.05°$ | N/A |
| Resolution [km] | $5.5 \times 5.5$ | 6 | $5.5 \times 5.5$ | 6 | $5.5 \times 5.5$ | $5.5 \times 5.5$ | 7 |
| Grid Size | $200 \times 200$ | N/A | $200 \times 200$ | N/A | $200 \times 200$ | $200 \times 200$ | N/A |
| Num. Datapoints | ~14.6M | ~205K | ~14.6M | ~955K | ~14.6M | ~14.6M | ~1.79M |
| Period Start | 2012-10-01 | 2012-10-01 | 2012-10-01 | 2012-10-01 | 2012-10-01 | 2012-10-01 | 2016-12-01 |
| Period End | 2013-09-30 | 2013-09-30 | 2013-09-30 | 2013-09-30 | 2013-09-30 | 2013-09-30 | 2018-01-31 |
| Frequency | Daily | 1 Hz | Daily | 1 Hz | Daily | Daily | 1 Hz |
| Period Length | 365 Days | 365 Days | 365 Days | 365 Days | 365 Days | 365 Days | 427 Days |
| Evaluation Start | 2012-10-22 | 2012-10-22 | 2012-10-22 | 2012-10-22 | 2012-10-22 | 2012-10-22 | 2017-01-01 |
| Evaluation End | 2012-12-02 | 2012-12-02 | 2012-12-02 | 2012-12-02 | 2012-12-02 | 2012-12-02 | 2017-12-31 |
| Evaluation Length | 45 Days | 45 Days | 45 Days | 45 Days | 45 Days | 45 Days | 365 Days |

# B  Physical Variables

As alluded to in the main body of the paper, we have access to many physical quantities which can be derived from sea surface height. This gives us a way to analyze how effective and trustworthy are our reconstructions. Many machine learning methods are unconstrained so they may provide solutions that are physically inconsistent and visualizing the field is a very easy eye test to assess the validity. In addition to post analysis, one could include some of these derived quantities maybe useful as additional inputs to the system and/or constraints to the loss function. Recall the spatiotemporal coordinates from equation 1, we use the same coordinates for the subsequent physical quantities. **Sea Surface Height** is the deviation of the height of the ocean surface from the geoid of the Earth. We can define it as:

$$\text{Sea Surface Height } [m]: \qquad \eta = \boldsymbol{\eta}(\mathbf{x}, t) \qquad \Omega \times \mathcal{T} \to \mathbb{R} \qquad (9)$$

This quantity is the actual value that is given from the satellite altimeters and is presented in the products for SSH maps [95]. An example can be seen in the first row of figure 4.

**Sea Surface Anomaly** is the anomaly wrt to the spatial mean which is defined by

$$\text{Sea Level Anomaly } [m]: \qquad \bar{\eta} = \boldsymbol{\eta}(\mathbf{x}, t) - \bar{\eta}(t) \qquad \Omega \times \mathcal{T} \to \mathbb{R} \qquad (10)$$

where $\bar{\eta}(t)$ is the spatial average of the field at each time step. An example can be seen in the first row of figure 2.

Another important quantity is the **geostrophic velocities** in the zonal and meridional directions. This is given by

$$\text{Zonal Velocity} [ms^{-2}]: \qquad u = -\frac{g}{f_0}\frac{\partial \eta}{\partial y} \qquad \Omega \times \mathcal{T} \to \mathbb{R} \qquad (11)$$

$$\text{Meridional Velocity} [ms^{-2}]: \qquad v = \frac{g}{f_0}\frac{\partial \eta}{\partial x} \qquad \Omega \times \mathcal{T} \to \mathbb{R} \qquad (12)$$

where $g$ is the gravitational constant and $f_0$ is the mean Coriolis parameter. These quantities are important as they can be an related to the sea surface current. The geostrophic assumption is a very strong assumption however it can still be an important indicator variable. The **kinetic energy** is a way to summarize the (geostrophic) velocities as the total energy of the system. This is given by

$$KE = \frac{1}{2}\left(u^2 + v^2\right) \qquad (13)$$

An example can be seen in the second row of figure 4.

Another very important quantity is the *vorticity* which measures the spin and rotation of a fluid. In geophysical fluid dynamics, we use the **relative vorticity** which is the vorticity observed within at rotating frame. This is given by

$$\zeta = \frac{\partial v}{\partial x} - \frac{\partial u}{\partial y} \qquad (14)$$

An example can be seen in the third row of figure 4.

We can also use the **Enstrophy** to summarize the relative voriticty to measure the total contribution which is given by

$$E = \frac{1}{2}\zeta^2 \qquad (15)$$

The **Strain** is a measure of deformation of a fluid flow.

$$\sigma = \sqrt{\sigma_n^2 + \sigma_s^2} \qquad (16)$$

where $\sigma_n$ is the shear strain (aka the shearing deformation) and $\sigma_s$ is the normal strain (aka stretching deformation). An example can be seen in the fourth row of figure 4.

The **Okubo-Weiss Parameter** is high-order quantity which is a linear combination of the strain and the relative vorticity.

$$\sigma_{ow} = \sigma_n^2 + \sigma_s^2 - \zeta^2 \qquad (17)$$

This quantity is often used as a threshold for determining the location of Eddies in sea surface height and sea surface current fields [80, 109, 90].

# C  Metrics

There are many metrics that are standard within the ML community but unconvincing for many parts the geoscience community. Specifically, many of these standard scores do not capture the important optimization criteria in the scientific machine learning tasks. However, there is not consensus within domain-specific communities about the perfect metric which captures every aspect we are interested. Therefore, we should have a variety of scores from different perspectives to really assess the pros and cons of each method we wish to evaluate thoroughly. Below, we outline two sets of scores we use within this framework: skill scores and spectral scores.

## C.1  Skill Scores

We classify one set of metrics as *skill scores*. These are globally averaged metrics which tend to operate within the real space. Some examples include the root mean squared error (RMSE), the normalized root mean squared (nRMSE) error, and the nRMSE score. The RMSE metric can also be calculated w.r.t. the spatial domain, temporal domain or both. For example, figure 3 showcases the nRMSE score calculated only on the spatial domain and visualized for each time step.

$$\text{RMSE}: \qquad\qquad \text{RMSE}(\eta, \hat{\eta}) = ||\eta - \hat{\eta}||_2 \qquad\qquad (18)$$

$$\text{nRMSE}: \qquad\qquad \text{nRMSE}(\eta, \hat{\eta}) = \frac{\text{RMSE}(\eta, \hat{\eta})}{||\eta||_2} \qquad\qquad (19)$$

$$\text{nRMSE}_{\text{score}}: \qquad\qquad \text{nRMSE}_{\text{score}}(\eta, \hat{\eta}) = 1 - \text{nRMSE}(\eta, \hat{\eta}) \qquad\qquad (20)$$

However, we are not limited to just the standard MSE metrics. We can easily incorporate more higher-order statistics like the Centered Kernel Alignment (CKA) [65] or information theory metrics like mutual information (MI) [61, 70]. In addition, we could also utilize the same metrics in the frequency domain as is done in [96].

## C.2  Spectral Scores

Another class of scores that we use in `OceanBench` are the *spectral scores*. These scores are calculated within the spectral space via the wavenumber power spectral density (PSD). This provides a spatial-scale-dependent metric which is useful for identifying the largest and smallest scales that were resolved by the reconstruction map. In general, we use these to measure the expected energy at different spatiotemporal scales and we can also construct custom score functions which gives us a summary statistic for how well we reconstructed certain scales.

$$\text{PSD}: \qquad\qquad \text{PSD}(\eta) = \sum_{k_{min}}^{k_{max}} ||\mathcal{F}(\eta)||^2 \qquad\qquad (21)$$

$$\text{PSD}_{score}: \qquad\qquad \text{PSD}_{score}(\eta, \hat{\eta}) = 1 - \frac{\text{PSD}(\eta - \hat{\eta})}{\text{PSD}(\eta)} \qquad\qquad (22)$$

where $\mathcal{F}$ is the Fast Fourier Transformation (FFT). In our application, there are various ways to construct the PSD which depend on the FFT transformation. We denote the *space-time PSD* as $\lambda_{\mathbf{x}}$ which does the 2D FFT in the longitude and time direction, then takes the average over the latitude. We denote the *space-time PSD* as $\lambda_{\mathbf{t}}$ which does the 2D FFT in the longitude and latitude direction, then takes the average over the time. We denote the *isotropic PSD* as $\lambda_r$ which assumes a radial relationship in the spatial domain and then averages over the temporal domain. Lastly, we denote the standard PSD score as $\lambda_a$ which is the 1D FFT over a prescribed distance along the satellite track; this is what is done for the OSE NADIR experiment. We recognize that the FFT configurations are limited due to their global treatment of the spectral domain and we need more specialized metrics to handle the local scales. This opens the door to new metrics that handle such cases such as the Wavelet transformation [104].

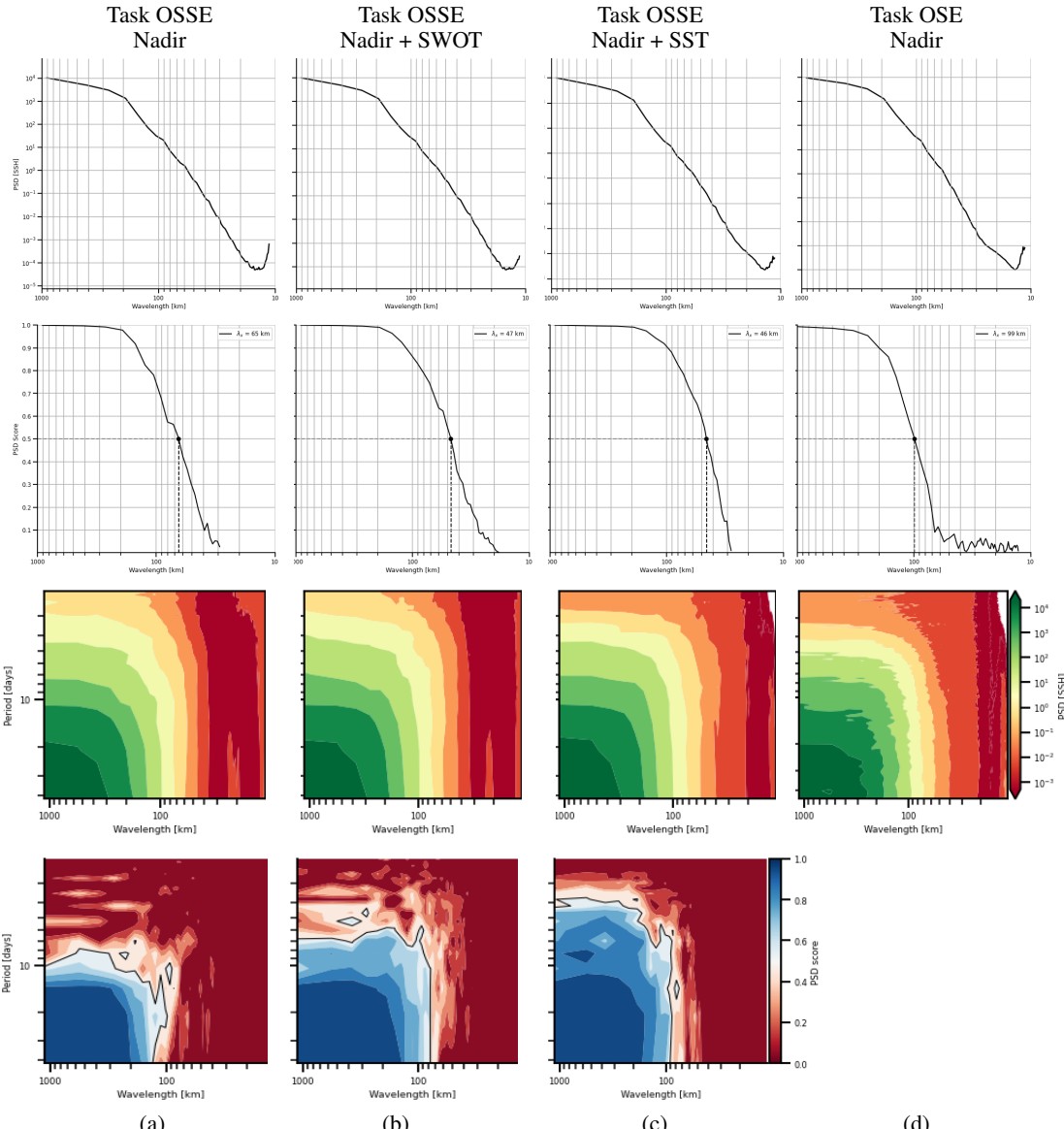

Figure 5: Power spectrum and associated scores of the 4dVarNet method for each of the four tasks. The row display in order: (1) the isotropic PSD, (2) the spatial PSD score (using the isotropic PSD for the first three rows and along track PSD for the last row), (3) the space-time PSD, (4) The spacetime PSD score available only in OSSE task.

# D   Use Case II: Hydra Recipes

This framework has drastically reduced the overhead for the ML researcher while also enhancing the reprducibility and replicability of the preprocessing steps. In this section we showcase a few examples for how one can use oceanbench in conjunction with hydra to provide recipes for some standard processes.

## D.1   Task Recipe

In this example, we showcase how we define an interpolation task for the OSE NADIR data challenge. We need to state the list of datasets available and specify which datasets are to be using for training and testings. We also specify the spatial region we would like to train on and the train-test period. There are a few simple changes one could do here to extend this task provided that one has uploaded standardized data that follows our set conventions. For example, for this interpolation task, the test period is a complete subset of the train period but one could imagine a forecasting task whereby the test period is at a completely different time period. Similarly, for this task, the train-test domain is the same but we could easily change the region of interest to see how the models perform in a completely different domain.

```
#@package _global_.task
outputs:
    # name of data challenge
    name: DC2021 OSE Gulfstream
    # list of datasets and locations
    data:
      train: # train data list
        alg: ${....data.outputs.alg}
        h2g: ${....data.outputs.h2g}
        j2g: ${....data.outputs.j2g}
        j2n: ${....data.outputs.j2n}
        j3: ${....data.outputs.j3}
        s3a: ${....data.outputs.s3a}
      test: # test data list
        c2: ${....data.outputs.c2}
    # spatial region specification
    domain: {lat: [33, 43], lon: [-65, -55]}
    # temporal period specification
    splits: {
        test: ['2017-01-01', '2017-12-31'],
        train: ['2016-12-01', '2018-01-31']
    }
```

Listing 1: This is a .yaml which showcases how we can communicate with Hydra framework to list a predefined set of specifications for the spatial region and the temporal period. This is an interpolation task for the OSE NADIR data challenge listed in section A.3.

## D.2 GeoProcessing Recipe

In this example, we showcase how one can pipe a sequential transformation through the hydra framework. In this example, we open the dataset, validate the coordinates to comply to our standards, select the region of interest, subset the data, regrid the alongtrack data to a uniform grid, and save the data to a netcdf file. See the listing D.2 for more information.

```yaml
# Target Function to initialize
_target_: "oceanbench._src.dataset.pipe"
# netcdf file to be loaded
inp: "${data_directory}/nadir_tracks.nc"
# sequential transformations to be applied
fns:
    # Load Dataset
    - {_target_: "xarray.open_dataset", _partial_: True}
    # Validate LatLonTime Coordinates
    - {_target_: "oceanbench.validate_latlon", _partial_: True}
    - {_target_: "oceanbench.validate_time", _partial_: True}
    # Select Specific Region (Spatial | Temporal)
    - {_target_: "xarray.Dataset.sel", args: ${domain}, _partial_: True}
    # Take Subset of Data
    - {_target_: "oceanbench.subset", num_samples: 1500, _partial_: True}
    # Regridding (AlongTrack -> Uniform Grid)
    - {
        _target_: "oceanbench.regrid",
        target_grid: ${grid.high_res},
        _partial_: True
    }
    # Save Dataset
    - {
        _target_: "xarray.Dataset.to_netcdf",
        save_name: "demo.nc",
        _partial_: True
    }
```

Listing 2: This is a `.yaml` which showcases how we can communicate with `Hydra` framework to list a predefined set of transformations to be *piped* through sequentiall. In this example, we showcase some standard pre-processing strategies to be saved to another netcdf file.

### D.3 Evaluation Recipe - OSSE

In this example, we showcase how one can use hydra to do the evaluation procedure. This is the same evaluation procedure that is used to evaluate the effectiveness of the OSSE NADIR experiment. From code snippet D.2, we see that we choose which target function to initialize and we choose the data directory where the .netcdf file is located. Then, we pipe some transformations for the .netcdf file: 1) validate the spatiotemporal coordinates, 2) we select the evaluation region, 3) we regrid it to the target get, 4) we fill in the nans with a Gauss-Seidel procedure, 5) we rescale the coordinates to be in meters and days, and 6) we perform the isotropic power spectrum transformation to get the effective resolution outlined in section C.

```yaml
# Target Function to initialize
_target_: "oceanbench._src.dataset.pipe"
# netcdf file to be loaded
inp: "${data_directory}/ml_result.nc"
# sequential transformations to be applied
fns:
    # Load Dataset
    - {_target_: "xarray.open_dataset", _partial_: True}
    # Validate LatLonTime Coordinates
    - {_target_: "oceanbench.validate_latlon", _partial_: True}
    - {_target_: "oceanbench.validate_time", _partial_: True}
    # Select Specific Region (Spatial | Temporal)
    - {_target_: "xarray.Dataset.sel", args: ${domain}, _partial_: True}
    # Regridding (Uniform Grid -> Uniform Grid)
    - {_target_: "oceanbench.regrid",
       target_grid: ${grid.reference}, _partial_: True}
    # Fill NANS (around the corners)
    - {_target_: "oceanbench.fill_nans",
       method: "gauss_seidel", _partial_: True}
    # Coordinate Change (degree -> meters, ns -> days)
    - {_target_: "oceanbench.latlon_deg2m", _partial_: True}
    - {_target_: "oceanbench.time_rescale",
       freq: 1, unit: "days", _partial_: True}
    # Calculate Isotropic Power Spectrum
    - {_target_: "oceanbench.power_spectrum_isotropic",
       reference: ${grid.reference}, _partial_: True}
    # Calculate Resolved Spatial Scale
    - {_target_: "oceanbench.resolved_scale", _partial_: True}
    # Save Dataset
    - {_target_: "xarray.Dataset.to_netcdf",
       save_name: "ml_result_psd.nc", _partial_: True}
```

Listing 3: This is a .yaml which showcases how we can communicate with Hydra framework to list a predefined set of transformations to be *piped* through sequential. In this example, we showcase some standard pre-processing strategies to be saved to another netcdf file.

# E   Use Case III: XRPatcher

There are many usecases for the `XRPatcher`. For example, we can do 1D Time chunking, 2D Spatial-Temporal Patches, or 3D Spatial-Temporal Cubes.

```python
import xarray as xr
import torch
import itertools
from oceanbench import XRPatcher
# Easy Integration with PyTorch Datasets (and DataLoaders)
class XRTorchDataset(torch.utils.data.Dataset):
    def __init__(self, batcher: XRPatcher, item_postpro=None):
        self.batcher = batcher
        self.postpro = item_postpro
    def __getitem__(self, idx: int) -> torch.Tensor:
        item = self.batcher[idx].load().values
        if self.postpro:
            item = self.postpro(item)
        return item
    def reconstruct_from_batches(
            self, batches: list(torch.Tensor), **rec_kws
        ) -> xr.Dataset:
        return self.batcher.reconstruct(
            [*itertools.chain(*batches)], **rec_kws
        )
    def __len__(self) -> int:
        return len(self.batcher)
# load demo dataset
data = xr.tutorial.load_dataset("eraint_uvz")
# Instantiate the patching logic for training
patches = dict(longitude=30, latitude=30)
train_patcher = XRPatcher(
    da=data,
    patches=patches,
    strides=patches,        # No Overlap
    check_full_scan=True    # check no extra dimensions
)
# Instantiate the patching logic for testing
patches = dict(longitude=30, latitude=30)
strides = dict(longitude=5, latitude=5)
test_patcher = XRPatcher(
    da=data,
    patches=patches,
    strides=strides,        # Overlap
    check_full_scan=True    # check no extra dimensions
)
# instantiate PyTorch DataSet
train_ds = XRTorchDataset(train_patcher, item_postpro=TrainingItem._make)
test_ds = XRTorchDataset(test_patcher, item_postpro=TrainingItem._make)
# instantiate PyTorch DataLoader
train_dl = torch.utils.data.DataLoader(train_ds, batch_size=4, shuffle=False)
test_dl = torch.utils.data.DataLoader(test_ds, batch_size=4, shuffle=False)
```

Listing 4: This is a snippet showcasing how we can easily integrate PyTorch Datasets within the `XRPatcher` framework without much overhead. Here we define a custom PyTorch Dataset to handle the `XRPatcher`. We load an arbitrary dataset with `xarray`, then we instantiate the `XRPatcher` with some patching logic, then we instantiate the PyTorch dataset and dataloader as per usual.

# F  Additional Tasks

In the main paper, we thoroughly outlined the interpolation task to showcase how `OceanBench` can be used to create automated pipelines for processing and evaluation procedures. However, there are many other additional tasks that can make use of the `OceanBench` features.

**Denoising**. A simpler problem for interpolation tasks is the denoising problem [99, 98]. The SSH and SST measurements we obtain have inherent noise from the sensors. A key problem is to calibrate the observations by separating the known noise patterns and the true signal. There has already been a lot of work from the ML side ranging from amortized predictions [100] to end-to-end learning schemes [46]. Much of this work has been facilitated by the *Ocean-Data-Challenge* group which have a few data challenges related to the denoising problem. Just like `OceanBench` was able to create reproducible pipelines from the SSH interpolation challenge listed in section 4.4, we also believe that one could extend the denoising challenge in the same manner.

**Forecasting**. This is a special form of extrapolation whereby the temporal domain of the state variable is sufficiently outside of the domain of the observation domain. Many previous benchmarking suites already look at forecasting for weather [86] and climate [108]. However, in oceanography, it is also advantageous to do forecasting for problems involving currents [91, 43] and eddies [76, 80, 90]. The `xrpatcher` will work out of the box for forecasting problems and contributions can be made to `OceanBench` to include some specific metrics for forecasting as were outlined in [86, 108, 10].

**Proxy Variables**. There are many other control variables that one could use to improve the interpolation or extrapolation task. We mentioned SST in section 4.4 because it is the most abundant observations available. However, there are other important observed variables which could be useful, e.g. Ocean colour, Bigeochemical parameters, and atmospheric variables. In many other downstream applications, the oceanography community often uses SSH and SST as proxy variables to predict important quantities related to the carbon uptake, e.g. SOCAT [11]. It would be straightforward to include a specific variable (and the associated preprocessing operations) into `OceanBench`.

**Dimension Reduction**. We often have very resolution spatiotemporal fields. which poses a very big challenge for learning due to the high correlations exhibited by spatiotemporal data and high dimensionality. A workaround for this is to learn a latent representation which retains as much relevant information as possible for the given task. In the ocean sciences, this is known as *Reduced Order Modeling* (ROM) or more generally dimensionality reduction which has been frequently used for adaptive meshes for physical models [117]. This could be used for pretraining fields to latent embeddings which could be useful for downstream tasks like anomaly detection [75].

**Surrogate Modeling**. Physical model simulations are very expensive and ML has played a role in learning surrogate models to descrease the computational intensity [93, 114]. We have a decently long spatiotemporal field over a region of interest which could be used to learn a surrogate model to mimic the dynamics of that region. This is also very useful for hybrid schemes whereby we have parameterizations to account for processes that are missing from low resolution simulations. [17, 54, 88, 57].

# G  Machine Learning Method Ontology

Although this paper does not focus on the explicit methods used for SSH interpolation, we would like to give a readers a brief overview of some of the most popular methods in the literature.

## G.1  Coordinate-Based methods

These methods learn a direct mapping between the coordinate vectors to the scalar or vector values.

$$\boldsymbol{y}_{obs} = \boldsymbol{f}(\mathbf{x}, t; \boldsymbol{\theta}) + \boldsymbol{\epsilon}(\mathbf{x}, t) \tag{23}$$

This is better known as *functa* [41] which parameterizes the field directly as a model.

**Functional**. Optimal Interpolation (OI) is the most common method used for many of the operational methods [95]. It is a non-parametric, functional method which is built upon covariance and precision matrices. In the machine learning community, these methods are known as Gaussian Process [72] and in the geostatistics community, this is known as Kriging [113].

**Basis Function**. This is an easy simplification to the functional by introducing parametric basis functions. In particular, the MIOST [102] algorithm will be adopted in the new operational products for SSH interpolation. It is a custom basis function based on Wavelet analysis which is scale-aware and scalable.

**Neural Fields**. Neural fields (NerFs) are a very popular set of methods that use neural networks to effectively learn the basis function through a composition of weights, biases and activations [62]. Furthermore, one can add physics-informed constraints to the loss function which mirror that of a PDE [63]. In many cases, especially with many auxillary inputs, we don't have access the PDE so one fit a NN directly to the observations with a fully connected neural network [11].

## G.2  Grid-Based Methods

In practice, we often consider the field at a specific discretized setting like a uniform grid or mesh. This is because we typically operate on and store these fields as multi-dimensional arrays which are only defined on a subspace of the entire continuous domain. We denote a discretized spatial representation as $\boldsymbol{\Omega}_g \subset \mathbb{R}^{N_s}$. We can simplify this notation by including the domain within the operator. So equation 7 like so:

$$\boldsymbol{\eta}(\boldsymbol{\Omega}_{obs}, t) = \mathcal{H}\left(\boldsymbol{\eta}(\boldsymbol{\Omega}_g, t), t, \boldsymbol{\mu}, \boldsymbol{\varepsilon}\right) \tag{24}$$

In this equation, $\mathcal{H}$ is the observation operator that transforms the field from the full discretized domain, $\boldsymbol{\Omega}_g$, to the observation domain, $\boldsymbol{\Omega}_{obs} \subset \mathbb{R}^{N_{obs}}$.

**Direct Methods**. These methods take the noisy, incomplete observations and directly feed it to a model that returns the full reconstructed field. They typically involve training a convolutional neural network or recurrent neural network on pairs of corrupted observations to learn the reconstruction [107, 81, 59]. This has seem some sucess in applications related to SSH interpolation [9, 74, 116].

**Traditional Data Assimilation.** There are many traditional methods that are rooted in data assimilation [21]. For example, the GLORYS [60] method propagates the physical model forwards in time and then *updates* the state based on observations periodically. A simpler approach is to use a nudging scheme coupled with a simpler physical model [55].

**End-to-End Learning**. These methods try to solve the problem by learning and end-to-end scheme to solve the model inversion problem. This is very similar to implicit methods that define a cost function to minimize instead of a minimizing the parameters of a prior model. Plug-in-Play priors are a popular class of methods that pre-train priors on auxillary observations and then use the prior in the inversion scheme [115]. This has seen a lot of success in SSH interpolation [14, 44, 43].

# H Limitations

## H.1 Framework Limitations

While we have advertised `OceanBench` as a unifying framework that provides standardized processing steps that comply with domain-expert standards, we also highlight some potential limitations that could hinder its adoption for the wider community.

**Data Serving**. We provide a few datasets but we omit some of the original simulations. We found that the original simulations are terabytes/petabytes of data which becomes infeasible for most modest users (even with adequate CPU resources). This is very big problem and if we want to have a bigger impact, we may need to do more close collaborations with specified platforms like the Marine Data Store [36, 33, 34, 31, 37, 32, 35] or the Climate Data Store [25, 23, 22, 24]. Furthermore, there are many people that will not be able to do a lot of heavy duty research which indirectly favours institutions with adequate resources and marginalizing others. This is also problematic as those communities tend to be the ones who need the most support from the products of such frameworks. We hope that leaving this open-source at least ensure that the knowledge is public.

**Framework Dependence**. The user has to "buy-into" the `hydra` framework to really take advantage of `OceanBench`. This adds a layer of abstraction and a new tool to learn. However, we designed the project so that high level usage does not require in-depth knowledge of the framework. In addition, we hope that, despite the complexity of project, users will appreciate the flexibility and extensibility of this framework.

**Lack of Metrics**. We do not provide the most exhaustive list of metrics available with the ocean community. In fact, we also believe that many of these metrics are often poor and do not effectively assess the goodness of our reconstructions. However, we do provide a platform that will hopefully be useful and easy to implement new and improved metrics. Furthermore, having a wide range of metrics that are trusted across communities may help to improve the overall assessment of the different model performances [50].

**Limited ML Scope**. The framework does not support nor promote any machine learning methods and we lack any indication of comparing ML training and inference performance. However, we argue that a benchmark framework will allow us to effectively compare whichever ML methods are demonstratively the best which is a necessary preliminary step which offers users more flexibility in the long-run.

**Broad Oceans Application Scope**. We have targeted a broad ocean-application scope of state estimation. However, there may be more urgent applications such as maritime monitoring, object tracking, and general ocean health. However, we feel that many downstream applications require high-quality maps. In addition, those downstream applications tend to be very complicated and are not always straightforward to apply ML under those instances.

**Full Pipeline Transparency**. We use a lot of different `xarray`-specific packages which have different design principles, assumptions and implementations. This may give the users an illusion of simplicity and transparency to real-world use. However, there are many underlying assumptions within each of the packages that may occlude a lot of design decisions. Despite this limitation, we believe that being transparent about the processing steps and being consistent with the evaluation procedure will be beneficial for the ML research community.

**Scalability**. Scaling this to many terabytes or petabytes of data is easily the biggest limitation of the framework. In addition, we have only showcased demonstrations for 2D+T fields which are much less expensive than 3D+T fields.

**Deployability**. MLOPs has many wheels and it is not easy to integrate into existing systems. We offer no solutions to this. However, we believe that our framework is fully transparent in the assumptions and use cases which will facilitate some adoption into operational systems where they can further modify it for their use cases (see the evolution of `WeatherBench` and `ClimateBench`).

**Visualization Tools**. We do not incorporate a high quality visualization tool that allows users to do pre- and post-analysis at a large scale. We do provide some simple visualization steps that are ML-relevant (see the GitHub repo) but it is very limited to ML standards. One solution is to interface our pipeline with the source of many ocean datasets, e.g. Climate Data Store [25] or Marine Data Store [36], then we can offset this task to them where they can offer better quality visualization tools.

## H.2    Data Challenge Limitations

We have showcased the SSH interpolation edition as a data challenge which could be helpful for real applications. However, in section 3.3 we alluded to the greater task of general ocean state estimation which is more pertinent to the ocean sciences yet we don't address this directly with our data challenges. Furthermore, we claim that the data challenges presented will help the ocean community with using ML for SSH interpolation. Below, we outline some limitations which address these criticisms.

**Not the overall objective**. We recognized that we are far away from the actual reanalysis and forecasting goals of full state estimation. However, we argue that that is a rather ambitious challenge which will require a lot of interdisciplinary work across communities. In the meantime while we work towards that goal, operational centers could possibly improve their current products from ML-based techniques would would benefit downstream applications that deal directly with SSH. Furthermore, SSH is an important variable in describing the full ocean state. So a robust set of techniques that are able to solve the interpolation tasks could (in principal) be used to solve extra tasks.

**Small Region & Period**. We only feature a small region and period over the Gulfstream which is not representative of the different global regimes. This also does not take into account real things like *data drift* which will inevitable occur in operational settings. However, this is a dynamical regime and a well-studied area which does have some importance for specific communities and the results obtained offer some transferability to other dynamical regimes. In addition, this area will have good coverage due to the new SWOT mission [51] which will allow for further validation in the future. Lastly, the area is small enough where the beginning stages for ML researchers is not overwhelmed with problems involving scale (even though we eventually want to arrive at global schemes). We hope to extend our challenges to more relevant scenarios [32].

**Simulations versus Reanalysis**. We use simulations for the OSSE experiments instead of reanalysis. This is an open research question as it is unclear whether it's better to pretrain models on simulated ocean data or reanalysis ocean data. In future updates, we plan to add the reanalysis data to extend the challenge.

**Efficacy of OSSE Experiments**. We alluded to the idea that the OSSE experiments may not reflect the overarching goal of the user yet we provide more OSSE experiments than OSE experiments. We acknowledged that it often does not coincide exactly with the OSE experiments which may give users a false sense of accomplishment and immediate transferability. However, we try to provide a framework where one could thoroughly experiment with the learning problem on OSSE configurations which can facilitate transfer learning to other domain-specific tasks. We also anticipate that new *real* SWOT data [51] will start to become more available which will allow us to design better, realistic OSE experiments.

**Noise Characterization**. Real data has noise to content with and we do not account for that within the SSH interpolation experiments. The true noise we see in operational settings is structured and this would require more knowledge outside the scope of our teams expertise. A more improved challenge would take these considerations into account. We leave this as a future challenge for the community and we hope our platform can help facilitate this.

**Uncertainty Quantitification**. We prefaced the problem statement with the idea of data assimilation which is the notion of *state/parameter estimation under uncertain conditions and incomplete information* [21]. However, we have not addressed any notion of uncertainty at all throughout the paper. Uncertainty is difficult to quantify and we don't want to impose too many restrictions until we more sure about the efficacy of ML for easier problems. However, to move the problem setting towards a more realistic setting, we can start to introduce metrics and additional requirements from future challenges, e.g. mean and standard deviation estimates or ensemble predictions.

**Operational Constraints**. The real use case of SSH interpolation will involve global data and/or high-resolution data. This involves dealing with very high-dimensional spatiotemporal global state-space. In practice, the necessity for the scalability of the method is of paramount importance. However, there are also areas within the ML research community who are looking into many ways we can scale up physical models [56, 85] and machine learn models for geoscience tasks [19]. We anticipate that once a set of solutions are excepted by a community, the scalability will come later.