# OpenReview forum: "OceanBench: The Sea Surface Height Edition"
_NeurIPS.cc/2023/Track/Datasets_and_Benchmarks — NeurIPS 2023 Datasets and Benchmarks Poster_

### Official Review · Reviewer_t7YW · 2023-07-21
**A workflow to move various SSH data to ML-ready state**

**Rating:** 7
**Confidence:** 3
**Correctness:** The dataset are construct in a reason…

**Strengths:**

OceanBench comprises a comprehensive set of materials, including data, code, and Jupyter notebook tutorials, specifically designed to facilitate ease of use for researchers. The introduction of these materials is presented in a detailed and thorough manner.

**Additional Feedback:**

The paper contains numerous acronyms that lack their full names.

In section 4.2, the statement "From a ML point of view, we can explore various ways to define the operator in equation (30)." references a nonexistent equation 30.

Additionally, in the caption of Figure 2, "PSD" appears before its full name.

Overall, it is a good paper.

**Clarity:**

The clarity of information presented in this paper is evident.






**Documentation:**

Accessing the data and code is effortless and straightforward.


**Ethics:**

There are no ethics concerns.

**Limitations:**

There is no negative social impact of this work.

**Opportunities For Improvement:**

Does utilizing data processed through OceanBench result in improved machine learning performance compared to ML trained with raw data?

Have you explored the possibility of introducing noise in the OSSE test?

Given that extrapolation tasks are typically more challenging than interpolation, I am interested in understanding how OceanBench performs on extrapolation tasks.

Has there been testing to assess the generalization capabilities between OSSE and OSE data? Can an OSSE trained model be effectively utilized on OSE data?

**Relation To Prior Work:**

Until now, there have been no previous attempts to unify various data into the same ML-ready format. This paper marks a commendable beginning in that direction.

**Summary And Contributions:**

The author presents OceanBench, an integrated and flexible platform designed to simplify the conversion of raw data into a machine-learning-ready format and translate model results into easily understandable metrics. This tool equips marine science researchers with readily available data, pre-configured pipelines, and evaluation instruments to address various challenges, including SSH interpolation problems.

---

> ### Author Response · Authors · 2023-08-11
> **Thank you for your review! Some clarifications.**
>
>
> >* Does utilizing data processed through OceanBench result in improved machine learning performance compared to ML trained with raw data?*
>
> On the one hand, we do not expect any improvement in the pure benchmarking performances we see in individual papers. However, the uptake of these models into other research tasks or operational settings will be easier because we have a fair and transparent platform that documents all decisions. In addition, there may be some ML-related improvements to preprocessing the heterogeneous data.
>
> ---
>
> > *Have you explored the possibility of introducing noise in the OSSE test?*
>
> We have considered it as an addition to the benchmark stack. Adding a processing step that would artificially add noise would be very easy. However, to ensure that it is relevant to the observations we expect from satellite data, we want to ensure that the noise added is consistent with actual satellite measurements. There are a few datasets in the Marine Data Store that have some noise characteristics shown, e.g. [AlongTrack](https://data.marine.copernicus.eu/product/SEALEVEL_GLO_PHY_L3_NRT_OBSERVATIONS_008_044/description).
>
> There is already a similar benchmark dataset available from the [ocean-data-challenge](https://github.com/ocean-data-challenges/2022a_SWOT_karin_error_filtering) group that looks at assessing denoising algorithms for an upcoming satellite mission. Adding this dataset to the platform and the respective preprocessing algorithms and metrics associated with the challenge would be easy.
>
> ---
>
> > *Given that extrapolation tasks are typically more challenging than interpolation, I am interested in understanding how OceanBench performs on extrapolation tasks.*
>
> We believe OceanBench would be well suited for forecasting tasks because changing the region, period, and train/test split is easy. From the platform's perspective, we need standard preprocessing tools and adequate metrics. However, the experimental design is a challenge for forecasting problems in the ocean. We do not have much "ground truth" available. In addition, benchmarking the predictive power of forecasting algorithms on reanalysis data for the ocean is also problematic because the uncertainty between the observations and model error is often poorly characterized. The good thing is that we have access to many observation data (albeit sparse), which can be used to validate the performance of forecasting models. We are interested in exploring ways to achieve this in upcoming editions.
>
> ---
>
> > *Has there been testing to assess the generalization capabilities between OSSE and OSE data? Can an OSSE trained model be effectively utilized on OSE data?*
>
> This is a known but relatively unexplored research direction. Like ML applied to other applications, it is well-known that models trained on OSSE (model simulations or reanalysis data) do not directly generalize to OSE (actual data) and require recalibration, bias correction, or parameter tuning. We hypothesize that it is necessary to design OSSE experiments because we don't have access to a lot of "ground truth" OSE data at the resolution and coverage that is necessary to train effective ML models. However, we are open to the idea of creating a challenge that tries to compare the performance of state estimation using only simulation data, reanalysis data, observation data, or some combination.
>
> ---
>
> > * *In section 4.2, the statement "From a ML point of view, we can explore various ways to define the operator in equation (30)." references a nonexistent equation 30.*
> > * *Additionally, in the caption of Figure 2, "PSD" appears before its full name.*
>
> We thank the reviewer for catching these errors. We have fixed them in the paper.

---

> > ### Comment · Reviewer_t7YW · 2023-08-30
> >
> > Thank you for answering all my questions! Looking forward to seeing the future work in extrapolation. I would like to keep my score.

---

### Official Review · Reviewer_DJRc · 2023-07-22
**Interesting Dataset but need more details on the benchmark**

**Rating:** 6
**Confidence:** 2

**Strengths:**

The paper describes the challenges ML researchers faced in processing

**Additional Feedback:**

I am hoping for more details on the framework in particular the details of the dataset used provided by the framework. The appendix contains some details on the spatial-temporal size and resolution OSSE Nadir, however there are no additional details provided for OSSE SSH, and OSE Nadir. It would also be good to summarize the datasets in the registry somewhere in the main paper.

Also, am I misinterpreting the table 2 in the appendix? The results with the larger normalized RMSE being bolded. Usually larger RMSEs implies worse performing models (4DVarNet).

**Clarity:**

The paper is somewhat well written, however because I have very little experience in oceanic science and dynamic systems, there are portions that I am unable to follow.

**Correctness:**

The authors mention that the complete list of the challenge of the benchmarks is in Table 3 of the appendix, however the table is not found in the appendix. I am unable to assess the correctness of the claims.

**Documentation:**

No, I was expecting and hoping for more details on the data, including a summary of the data provided by third parties.

**Ethics:**

No, I do not suspect there are any ethical concerns.

**Limitations:**

Yes, the authors addressed the limitations of their work

**Opportunities For Improvement:**

I am hoping for more details on the framework in particular the details of the dataset used provided by the framework. The appendix contains some details on the spatial-temporal size and resolution OSSE Nadir, however there are no additional details provided for OSSE SSH, and OSE Nadir. It would also be good to summarize the datasets in the registry somewhere in the main paper.

Also, am I misinterpreting the table 2 in the appendix? The results with the larger normalized RMSE being bolded. Usually larger RMSEs implies worse performing models (4DVarNet).

**Relation To Prior Work:**

Yes, the authors discussed other environmental and simulated benchmarks for dynamical systems. and how their work is tailored for oceanic modelling.

**Summary And Contributions:**

This paper describes OceanBench, a benchmarking framework for benchmarking ML models on ocean satellite data. This framework

---

> ### Author Response · Authors · 2023-08-11
>
> > *I am hoping for more details on the framework in particular the details of the dataset used provided by the framework. The appendix contains some details on the spatial-temporal size and resolution OSSE Nadir, however there are no additional details provided for OSSE SSH, and OSE Nadir. It would also be good to summarize the datasets in the registry somewhere in the main paper.*
>
> We acknowledge this mishap on our part. We should have included more details about the dataset, as it is vital for ML practitioners. We added a table (**Table 1, Section 4.3**) highlighting the most critical data characteristics to address this point. We have added a larger table (**Table 4, Appendix A.5**) to include more details like training and evaluation periods and alongtrack versus gridded dataset characteristics.
>
> ---
>
> > *Also, am I misinterpreting the table 2 in the appendix? The results with the larger normalized RMSE being bolded. Usually larger RMSEs implies worse performing models (4DVarNet).*
>
> We apologize for the confusion. We will rename this as the normalized RMSE *score*. The normalized RMSE score would be the normalized RMSE subtracted from 1 to get the *score* that is easily comparable.
>
> $$
> \\text{nRMSE}(\\eta,\\hat{\\eta}) = \\frac{\\text{RMSE}(\\eta,\\hat{\\eta})}{||\\eta||_2}
> $$
>
> $$
> \\text{nRMSE}_{\\text{score}}(\\eta,\\hat{\\eta}) = 1 - \\text{nRMSE}(\\eta,\\hat{\\eta})
> $$
>
>
> This is why the best results listed in Table 2 will have a nRMSE (score) closer to 1.0, which is the 4DVarNet method that was mentioned. To clarify this, we have renamed these to *nRMSE score* within all tables and the metrics sections. We have also included the above formula to provide more clarification for the reader.
>
> ---
>
> > *The authors mention that the complete list of the challenge of the benchmarks is in Table 3 of the appendix, however the table is not found in the appendix. I am unable to assess the correctness of the claims.*
>
> This was an error on our part. The tables have been fixed as follows:
> * **Table 2 - Section 4.5** - it has a summary of the results of the OSSE NADIR benchmark challenge
> * **Table 3 - Appendix A.4** - has the full table with the result for all four challenges.
>
> ---
>
> > *No, I was expecting and hoping for more details on the data, including a summary of the data provided by third parties.*
>
> To echo the statements above, this was an oversight, and we have corrected it by adding the two tables to highlight the data characteristics.

---

> > ### Comment · Reviewer_DJRc · 2023-08-30
> > **Thank you for the clarification**
> >
> > Thank you for the clarification. It does address some of my issues with the paper. I will adjust my ranking accordingly.

---

### Official Review · Reviewer_nYg2 · 2023-07-23
**Review of OceanBench**

**Rating:** 7
**Confidence:** 4
**Correctness:** To my knowledge, claims are correct.

**Strengths:**

- I appreciated that the authors aimed to provide a benchmark that was usable by ML researchers while still remaining relevant for domain scientists and real world applications (i.e., they didn't dilute the task to make it easier to use for benchmarking).
- I liked the attention to data-centric aspects of ML development, which as the authors pointed out are often more important than the model architecture or other model-centric choices for geoscience use cases.
- The authors made design choices (e.g. use of xarray) to ensure the benchmark was interoperable and remained domain relevant.

**Additional Feedback:**

It is not clear from the paper what the "unmet needs" are of current SSH methods and datasets that necessitate the creation of the proposed dataset. What improvements do the authors hope to catalyze with this dataset that is not done by the existing Ocean Data Challenge datasets? It would be helpful to add a short discussion of this to the paper.

**Clarity:**

The paper is overall clearly and easy to understand (beyond the lack of technical dataset details discussed earlier in this review).

**Documentation:**

There seems to be adequate documentation but details like the license should be more explicit. I did not see the license listed in the paper nor on the data registry github. The checklist says "The appropriate license notices are included in the affect source code files, and license of new assts is included in the supplementary materials." However, I did not see the license in the supplementary materials.

**Ethics:**

I did not see any ethical concerns.

**Limitations:**

- In general, I felt that the details in the paper are not sufficient for understanding the content of the dataset, how to use it, and what the impact of progress on the proposed benchmark would be (see previous comments about limitations).
- The authors did not discuss limitations of the dataset such as the geographic or temporal scope and how that might affect the use or impact of the dataset.

**Opportunities For Improvement:**

- At times the authors seem to be pitching OceanBench as a general purpose framework for hosting geoscience ML tasks, not a benchmark for SSH. For example, "the SSH edition" - what are the next editions? This is not discussed. It's not clear how the provided tools can be extended for other tasks or even different versions of the SSH task (different regions, time periods, etc). I think the proposal would be more clear if the authors focused on SSH, or describe in more detail how someone could extend OceanBench for other tasks/domains.
- The paper does not contain enough details about the actual dataset that is being proposed. The benchmark challenges are described, but the reader does not know critical things about the dataset like how large it is, what is the dimensionality, etc. A lot of space is used in the paper to describe background (like the problem definition) that I feel are overlong and not necessary for understanding the content of or how to use the dataset, which is important for ML practitioners.

**Relation To Prior Work:**

The discussion of related work could be improved by providing more discussion of existing ocean datasets. The current discussion seems to try to address geoscience datasets broadly, but then only discusses a small part of geoscience datasets. It would have been better to describe more ocean datasets and how the proposed datasets add to the current state of datasets more specifically than is done in the last sentence of Section 2.

**Summary And Contributions:**

The authors present OceanBench, a framework for benchmarking ML models using preprocessing and evaluation procedures developed with domain experts. The benchmark provides four benchmarking experiments that cover different types of challenges for estimating sea surface height (SSH). The authors reported baseline results from different types of methods commonly used to address each challenge.

---

> ### Author Response · Authors · 2023-08-11
> **Thank you for your review! Some clarifications [1/2]**
>
>
>
> > *At times the authors seem to be pitching OceanBench as a general purpose framework for hosting geoscience ML tasks, not a benchmark for SSH. For example, "the SSH edition" - what are the next editions? This is not discussed. It's not clear how the provided tools can be extended for other tasks or even different versions of the SSH task (different regions, time periods, etc). I think the proposal would be more clear if the authors focused on SSH, or describe in more detail how someone could extend OceanBench for other tasks/domains.*
>
>
> This is a great point and correctly highlighted the message we want to convey:
> * OceanBench is a general-purpose platform that can facilitate the experimental design of ocean-related applications (involving state estimation).
> * SSH interpolation is one instance of the (state estimation) problems for which OceanBench can facilitate the ML-friendly, experimental design.
>
> However, the scope of OceanBench could have been better described in the paper, and we could have sufficiently highlighted how it can be extended outside of the SSH interpolation use case that we demonstrated. To rectify this, we have added a section (**Section 3.3 - Problem Scope**) to summarize our beliefs about the scope of problems within the oceanography community that OceanBench can be useful for, i.e., problems under the state estimation umbrella. We have also included a few concrete example research questions that would interest operational and academic oceanographers alike.
>
> From an implementation perspective, if the dataset is available and we have added the appropriate preprocessing strategies necessary to standardize/homogenize it, then selecting the region, period, preprocessing strategy, or variables is very easy. This was done purposefully so that it is easy to work with heterogeneous observation datasets for different state estimation tasks, including interpolation and forecasting.
>
>
> We have added a small example of a *task* API in **Appendix D.1**. To demonstrate the customizability of the tasks via Oceanbench, we now showcase the following:
>
> * **Appendix D.1** shows how one can configure the region, period, and train/test split strategy (either temporally or spatially)
> * **Appendix D.2** shows how one can configure a custom preprocessing strategy which is dataset/task dependent
> * **Appendix D.4** shows how we can take an arbitrary gridded xarray data structure which can be ported to a PyTorch dataloader (also see [notebook](https://jejjohnson.github.io/oceanbench/content/getting_started/pytorch_dataset_integration_example.html))
> * **Appendix D.3** shows how one can configure a custom post-processing strategy for evaluation which is dataset/task-dependent.
> * We have a more [detailed example](https://jejjohnson.github.io/oceanbench/content/getting_started/TaskToPatcher.html) showcasing how we can generate PyTorch datasets via these task configurations.
>
>
> ---
>
> > *The paper does not contain enough details about the actual dataset that is being proposed. The benchmark challenges are described, but the reader does not know critical things about the dataset like how large it is, what is the dimensionality, etc. A lot of space is used in the paper to describe background (like the problem definition) that I feel are overlong and not necessary for understanding the content of or how to use the dataset, which is important for ML practitioners.*
> > ...
> > In general, I felt that the details in the paper are not sufficient for understanding the content of the dataset, how to use it, and what the impact of progress on the proposed benchmark would be (see previous comments about limitations).
>
>
> This is correct: We did not give the dataset content for the presented challenges its proper treatment, which is essential for ML practitioners and domain experts alike. We added a few tables highlighting essential data characteristics to address this point. The smaller table in the main paper (**Table 1, Section 4.3**) highlights the most essential data characteristics. We have added a larger table in the appendix (**Table 4, Appendix A.5**) to include more details like training and evaluation periods and alongtrack versus gridded dataset characteristics.
>
> Regarding the elongated problem definition section, we intended to appeal to applied ML researchers from geoscience and traditional ML backgrounds. From our experience, cross-disciplinary research always involves more time agreeing upon the problem definition and scope in addition to the available data structures and computational constraints. We tried hard to compromise between the two intended audience members to avoid isolating either community by glossing over implicit aspects of the problem definition. In addition, the framework's highlight is not the dataset: it is more about the experimental design aspects to encapsulate the essence of the problem one wishes to solve. And then, of course, OceanBench can help put it in an ML-ready format.

---

> ### Author Response · Authors · 2023-08-11
> **Thank you for your review! Some clarifications [2/2]**
>
> > *The authors did not discuss limitations of the dataset such as the geographic or temporal scope and how that might affect the use or impact of the dataset.*
>
> This is a great point: We should have highlighted *any* of the inherent limitations of the data challenges and dataset associated with the challenge. To address this, we have added a section in the appendix (**appendix H.2**) where we have highlighted many limitations in the challenge and data, including the comment about the region and temporal scope.
>
> ---
>
> > *The discussion of related work could be improved by providing more discussion of existing ocean datasets. The current discussion seems to try to address geoscience datasets broadly, but then only discusses a small part of geoscience datasets. It would have been better to describe more ocean datasets and how the proposed datasets add to the current state of datasets more specifically than is done in the last sentence of Section 2.*
>
> This is another great remark: We should have focused on ocean-related datasets relevant to our problem. To address this point, we have added a few more references with specific ocean-related datasets that we have come across, which are related to our problem definition described in the paper (**Section 2 - Related Work**). This will be more congruent with our intentions for OceanBench: we do not want to propose brand new datasets; we want to acknowledge the existence of available ocean-related datasets and integrate them into the benchmark framework, which helps demonstrate the usefulness of ML.
>
> ---
> > *There seems to be adequate documentation but details like the license should be more explicit. I did not see the license listed in the paper nor on the data registry github. The checklist says "The appropriate license notices are included in the affect source code files, and license of new assts is included in the supplementary materials." However, I did not see the license in the supplementary materials.*
>
> We thank the reviewer for catching this error. We have added the license to the [OceanBench-Data-Registry](https://github.com/quentinf00/oceanbench-data-registry) repo and also have added a phrase within the main paper about the details of the license (**Appendix A.5 - Datasets**). We have also updated the checklist to reflect this.

---

> ### Comment · Reviewer_nYg2 · 2023-08-26
> **Revised manuscript addresses major concerns**
>
> Thanks to the authors for considering my feedback in their revisions and thoroughly addressing the concerns. I feel that my concerns about the dataset have been addressed and I will raise my score.

---

### Official Review · Reviewer_fpi1 · 2023-07-31
**The paper introduces a framework tailored for handling challenging ocean interpolation and extrapolation tasks within the machine learning domain. However, to bolster the validity and reliability of the work, it is advisable to incorporate additional details about the background and domain knowledge of the data and scientific questions, with input from ocean domain experts.**

**Rating:** 6
**Confidence:** 4

**Strengths:**

(1)	Addressing an Important Problem: The paper addresses the challenges associated with ocean observation data, which is crucial for understanding the Earth's system and climate regulation. The introduction of the OceanBench framework offers significant benefits to researchers in both the machine learning and geoscience domains. The OceanBench framework has the potential to tackle complex geoscience open questions through the application of advanced machine-learning techniques. Simultaneously, it facilitates machine learning researchers in efficiently accessing research questions and relevant data for their studies.
(2)	Benchmarking and Reproducibility: With plug-and-play data and pre-configured pipelines, OceanBench enables ML researchers to benchmark their models against ML and domain-related baselines. This promotes fair comparisons and enhances the reproducibility of results, making it easier for researchers to validate their models.


**Additional Feedback:**

N/A

**Clarity:**

Yes, the paper is well-written overall. The authors have effectively communicated their research objectives and methodology in an organized manner. However, the language could be improved to make it easy to follow the flow of ideas.



**Correctness:**

Yes, the submitted framework has successfully provided general and domain-specific metrics for evaluating machine learning models. Additionally, it includes practical experiment examples that demonstrate how to utilize the proposed framework effectively. This comprehensive approach enhances the framework's applicability and usefulness for researchers in the machine learning domain.

**Documentation:**

The authors have provided a GitHub repository link with detailed information regarding the benchmark.  It allows researchers to access the benchmark's implementation and configuration for reproducibility.

**Limitations:**

The authors have acknowledged the framework's limitations in Section H of the appendix. In addition to these acknowledged limitations, it would be beneficial for the framework to include visualization tools that support analysis. By incorporating visualization tools, researchers would have a more comprehensive and intuitive platform to explore and interpret the processed data, model outputs, and evaluation metrics, thereby enhancing the overall usability of the OceanBench framework.

**Opportunities For Improvement:**

The paper does not provide a comprehensive introduction to the background of the challenges it aims to address. It may create difficulties for machine learning researchers in fully grasping the scientific questions and complexities associated with ocean satellite data, hindering their ability to develop meaningful solutions. Despite offering a valuable framework, there are still significant barriers for machine learning researchers to overcome in understanding the scientific context and challenges related to ocean data.



**Relation To Prior Work:**

The paper discusses differences from previous work in the motivation, related work, and objectives sections. However, a more comprehensive literature review could improve the related work section. The paper categorized related studies based on three data inputs: simulation, reanalysis, and observation data. However, it's worth noting that deep learning models in geoscience often handle similar data formats despite different data inputs.

**Summary And Contributions:**

The paper introduces OceanBench, a framework designed as a standardized benchmark for ML researchers. It offers pre-configured pipelines, data, and domain-related metrics to evaluate ML models while also allowing researchers to customize and extend the pipelines for their specific tasks. Specifically, it addresses SSH interpolation challenges in ocean satellite data, multi-modal and multi-sensor fusion, and transfer learning with ocean satellite observations.

---

> ### Author Response · Authors · 2023-08-11
> **Thank you for your review! Some clarifications.**
>
> > The paper does not provide a comprehensive introduction to the background of the challenges it aims to address. It may create difficulties for machine learning researchers in fully grasping the scientific questions and complexities associated with ocean satellite data, hindering their ability to develop meaningful solutions.
>
> This is an excellent point, as many ocean applications are outside the one we stated here. For our first edition, we wished to tackle a problem that:
> a. is beneficial for the ocean community
> b. is something ML researchers can tackle independently (given an adequate experimental platform).
> We chose the problem of state estimation of crucial ocean variables given observations, dynamical priors, and an initial state, which limits our scope to all problems formulated as state estimation problems. However, many problems in the oceanographic community can be formulated as a state estimation problem which could offer some knowledge to many other relevant tasks.
>
> To address this concern and clarify our intentions, we added a section (**section 3.3**) specifying the subclass of problems that OceanBench can help facilitate the experimental design for ML-ready tasks.
>
>
> > In addition to these acknowledged limitations, it would be beneficial for the framework to include visualization tools that support analysis. By incorporating visualization tools, researchers would have a more comprehensive and intuitive platform to explore and interpret the processed data, model outputs, and evaluation metrics, thereby enhancing the overall usability of the OceanBench framework.
>
> This is a great point. We acknowledge that we often need to revert to visualizations (pre and post-predictions) because often our metrics do not sufficiently capture some of the aspects we wish to preserve. While very important, a fully-fledged visualization tool is outside this project's scope due to our lack of expertise and budget constraints. However, we provide a few light scripts and demos so that users can do some simple plots to help facilitate analysis. For example:
>
> * simple visualizations of the observations, e.g. [alongtrack](https://jejjohnson.github.io/oceanbench/content/alongtrack/AlongTrack.html), [SWOT](https://jejjohnson.github.io/oceanbench/content/alongtrack/AlongTrack_SWOT.html)
> * simple maps of important [gradient-based derived variables](https://jejjohnson.github.io/oceanbench/content/geoprocess/physical_variables.html) that are very relevant quantities for oceanographers
> * [power spectrum](https://jejjohnson.github.io/oceanbench/content/metrics/power_spectrum.html) plots that are well-known for assessing the quality of simulations in the oceanographic community
> * simpler statistics based on [pixel densities](https://jejjohnson.github.io/oceanbench/content/viz/pixel_densities.html) which are very common in the data analysis community
> * and an [automated script](https://jejjohnson.github.io/oceanbench/content/getting_started/PlotMaps.html) that will plot the users and leaderboard results
>
> We want to highlight some work that exists, like the [Copernicus Marine Data Store](https://data.marine.copernicus.eu/products), which has an excellent [online visualization tool](https://marine.copernicus.eu/access-data/ocean-visualisation-tools) of all of their available datasets and [statistics about tracking datasets](https://pqd.mercator-ocean.fr/). We applaud this effort and hope that we can use their freely available datasets in our ML benchmark platform for future editions.
>
> To address this point, we have added this as a limitation in **appendix H.1**, highlighting this point and listing the above references.
>
>
> > The paper discusses differences from previous work in the motivation, related work, and objectives sections. However, a more comprehensive literature review could improve the related work section.
>
> We thank the reviewer for suggesting this. We have done another literature review and updated **section 2** with a more exhaustive list of other ocean-related datasets that are freely available. However, to the authors' knowledge, we have not found satisfactory references directly related to what we have tried to do - i.e., ML-ready datasets or benchmarks.
>
>
> > The paper categorized related studies based on three data inputs: simulation, reanalysis, and observation data. However, it's worth noting that deep learning models in geoscience often handle similar data formats despite different data inputs.
>
> We completely agree. This is why ML methods are worth investigating. However, combining heterogeneous datasets from different sources that are reproducible for benchmarking purposes and transparent for multi-disciplinary users is still a challenge. In the oceanography community, there are many techniques to homogenize the data that are either hidden or arbitrary. We hope this framework is a first step in opening the black box to these techniques. It could promote more ML-inspired methods to deal with heterogeneity.

---

### Author Response · Authors · 2023-08-11
**We thank the reviewers for their feedback! Our general comments to all reviewers [1/2]**

# General Comments to All Reviewers [1/2]

We thank the reviewers all very much for their reviews. The positive comments showed us that this framework can be very relevant to the ML community and impactful to the world of Ocean sciences. However, a few key, common criticisms appeared amongst the reviewers, and we would like to address those collectively in addition to individual comments.

---

## 1 - Lack of Dataset Details

A few reviewers mentioned that they were disappointed to see the lack of information about the datasets used within the challenges. This was a gross oversight on our part. We focused on many problems reformulating the geoscience problem from a ML perspective that appeals to domain experts and ML researchers. The problem formulation is where we have had the most difficulties in cross-disciplinary communication. However, we should have included essential details about the dataset, which is also very important when defining the data challenge to address our formulated problem. To rectify this, we added a small table in the main paper (**section 4.3**) to overview the datasets used in the demonstrated challenges. We have also added a complete table regarding some extra details about the experimental setup explained in the appendix (**appendix section A.5**). The tables highlight the dataset's most essential characteristics: the variable, region, spatial extent, spatial resolution, period, and frequency of simulation/observation outputs.

---

##  2 - Unaccounted for Limitations

As the reviewers pointed out, many limitations needed to be accounted for based on our decisions regarding the framework, the problem scope, and the data challenges introduced. To address this, we have extended the limitations section in the appendix with two separate sections: one outlines the limitations of the *OceanBench framework* itself, and the other outlines the limitations of the *SSH data challenges* proposed. While we hope the community will help us address many of the limitations in the future, there are still some limitations due to our framework and experimental design choices, which we should look to improve soon.

---

### Author Response · Authors · 2023-08-11
**We thank the reviewers for their feedback! Our general comments to all reviewers [2/2]**

# General Comments to All Reviewers [2/2]

## 3 - Clarity between OceanBench and the SSH Edition


Based on the comments, we needed to explicitly distinguish the **OceanBench** *platform* and the **SSH Edition** we discussed. We want to re-explain our perspective on how OceanBench is related to the SSH editions. We have added a section in the paper explaining the role of OceanBench in the ocean-related problems we wish to address (**section 3.3**). Below, we will paraphrase the paragraph within the paper.

**Overall Goal**. The problem relevant to domain scientists we want to tackle is to predict the ocean state given all of the available information, i.e., initial state, constraints, and observations. In addition to the scientific knowledge gained, it would greatly facilitate the improved production of high-quality data products, e.g. [Marine Data Store](https://data.marine.copernicus.eu/products), essential for many downstream [use cases](https://marine.copernicus.eu/services/use-cases). However, we encounter many issues that make this problem challenging, e.g., uncertainty in the priors, observations, and initial states. Furthermore, many logistical constraints further complicate the objective, e.g., high-dimensional data, multi-scale complexity, and an extreme amount of missing data. As such, the community is still in the early stages of designing an experiment within a ML framework that directly applies to the domain-specific research questions.

**SSH Edition**. Tackling the full ocean state estimation problem directly is difficult, but ML can tackle many intermediate problems, which are a gateway to the final objective. We focus on problems that can be posed as state estimation problems stemming from the original data assimilation strategies used within the ocean community for decades. Under this umbrella, we could address the following research questions that are important for the operational oceanography community:
* How can we effectively use heterogeneous observations to predict the ocean state on the sea surface? (*Interpolation Problem*)
* How can we incorporate prior knowledge into our predictions of ocean state trajectories? (*Data Assimilation*, *Bayesian Inference Problem*)
* How can we use the current ocean state now to predict the ocean state in the future? (*Forecasting Problem*)

There are also a few other research questions that would be of interest to the academic ocean modeling community:

* Is simulated or reanalysis data more effective for learning ML emulators that replace expensive ocean models?
* What metrics are more effective for assessing our ability to mimic ocean dynamics?
* How much model error can we characterize when learning from observations?

We chose SSH interpolation as one instance of the potential problems we hope to address in the Ocean community for state space estimation. SSH interpolation is a mature problem that is well-defined, and there are many observations we can use and operational products that we can compare with new ML methods. We have hypothesized that the oceanographic community has sufficient data and the ML community has sufficient expertise to incorporate machine learning effectively into these problems.

**OceanBench**. We designed this framework to help users create ML-oriented experiments focusing on handling heterogeneous, ocean-related datasets. On the front end of the full pipeline, it has a suite of processing tools that help to deal with heterogeneous observation data to make it "ML-ready" for consumption. On the back end of the full pipeline, it has a suite of domain-expert validated derived variables and metrics, which provide consistent assessment criteria for challenges. Overall, it is fairly agnostic and extensible to the types of ocean-related datasets, processing techniques, and metrics needed for working with a specific class of Ocean-related datasets. We hope it facilitates quality experimental designs with an easy API for ML consumption.



**OceanBench Demonstration**. This paper showcases how OceanBench can be tailored to configure an experimental platform for SSH interpolation (**section 4.3-4.5**). We created a fully-fledged pipeline that takes different heterogeneous datasets and preps them. OceanBench also takes results from previous research papers to make a leaderboard of metrics (**table 3**) and plot functionality (**Figure 3 & 4**). Although we only demonstrate it on SSH interpolation, this is the first step in other specific tasks, e.g., benchmarking emulators and forecasting models. The logistical differences between some of the problems, e.g., selection of a region, period, variables, observation sources, metrics, and train-test split, are all easily handled by the OceanBench platform (see **appendix D** for demo configurations).

---

### Decision · Program_Chairs · 2023-09-22

**Decision:**

Accept (Poster)

**Comment:**

The presented OceanBench framework provides a valuable tool that can enable geoscience research using ML techniques as well as ML research, and support reproducibility. The reviewers are positive about this submission and are largely satisfied with how the authors have addressed their concerns.